# An enzymatic cascade enables sensitive and specific proximity labeling proteomics in challenging biological systems

Tommy J. Sroka [1,2], Lea K. Sanwald [1,10], Avishek Prasai [1,10], Josefine Hoeren[1,3], Valentina Trivigno [3], Valerie Chaumet [1,2], Louisa M. Krauß [1], Damian Weber [4], Daniela Yildiz[2,5,6,7,8], Karina von der Malsburg [1], Peter Walentek [4], Per Haberkant[9], Bianca Schrul [1,6,8], Kerstin Feistel [3] & David U. Mick [1,2,6,8] ✉

Ascorbate peroxidase (APEX) is a proximity labeling enzyme used for sub-cellular proteomics at high spatial and temporal resolution. However, toxicity of its substrate hydrogen peroxide and background labeling by endogenous peroxidases limit its use to in vitro studies of specific cell types. To minimize toxicity and reduce non-specific background labeling we establish a more versatile in situ APEX activation (iAPEX) workflow by combining APEX2 with a D-amino acid oxidase to locally produce hydrogen peroxide. Using iAPEX, we profile the proteomes of a cellular microdomain, the primary cilium, in cell lines not readily accessible to conventional APEX labeling and identify unknown ciliary proteins. Our study validates common ciliary proteins across two distinct cell lines, while observed differences may reflect heterogeneity in primary cilia proteomes. Furthermore, iAPEX proximity labeling is applicable to a range of cellular compartments including mitochondria and lipid droplets and can be employed in *Xenopus laevis*, which provides a proof-of-concept for future in vivo applications.

Proximity labeling technologies provide the biological and chemical sciences with various applications to study nucleic acids, lipids and proteins[1–4]. Conceptually, proximity labeling uses enzymes fused to proteins of interest or directed to specific locations to determine their molecular environments by purifying proximity labeled biomolecules. Combining this approach with mass spectrometry-based proteomics allows unbiased and systematic determination of transient protein interactions as well as (sub)proteomes of organelles or cellular microcompartments that are inaccessible to biochemical purification methods[5,6].

The primary cilium is a solitary plasma membrane microdomain with important functions in developmental biology and tissue maintenance[7,8]. It functions as a specialized signaling compartment that translates extracellular cues into cellular responses by intricate mechanisms, employing second messengers and dynamic protein transport to and from the primary cilium[9,10]. Defects in these processes

[1]Center for Molecular Signaling (PZMS), Department of Medical Biochemistry and Molecular Biology, Saarland University School of Medicine, Homburg, Germany. [2]Center of Human and Molecular Biology (ZHMB), Saarland University School of Medicine, Homburg, Germany. [3]Department of Zoology, Institute of Biology, University of Hohenheim, Stuttgart, Germany. [4]Internal Medicine IV, Medical Center, CIBSS Centre for Integrative Biological Signalling Studies, SGBM Spemann Graduate School for Biology and Medicine, University of Freiburg, Freiburg, Germany. [5]Experimental and Clinical Pharmacology and Toxicology, Molecular Pharmacology, Center for Molecular Signaling (PZMS), Saarland University, Homburg, Germany. [6]PharmaScienceHub (PSH), Saarbrücken, Germany. [7]Center for Gender-specific Biology and Medicine (CGBM), Saarland University, Homburg, Germany. [8]Center for Biophysics (ZBP), Saarland University, Saarbrücken, Germany. [9]EMBL Heidelberg, Proteomics Core Facility, Heidelberg, Germany. [10]These authors contributed equally: Lea K. Sanwald, Avishek Prasai. ✉e-mail: david.mick@uks.eu

have been implicated in syndromic disorders, termed ciliopathies, affecting several tissues and cell types[7]. A current hypothesis posits that cell type-dependent differences in the composition of primary cilia account for the pleiotropy of these syndromic disorders, as cilia dysfunctions in specific signal transduction mechanisms may have cell type- and tissue-specific consequences. Due to its small size (~1:10,000th of the cell)[11] and difficulty to isolate pure primary cilia by classic biochemical methods[12], proximity labeling approaches have been utilized to determine primary cilia proteomes of disease models and to investigate basic cilia biology[13–16], such as dissecting the molecular composition of primary cilia during active signaling[17,18]. However, our knowledge of the protein composition of primary cilia is still incomplete as it stems from very few model cell types that are amenable to the available technologies.

While proximity labeling technologies are an active area of research[19], the most frequently used proximity labeling methods are based on two enzymatic activities that use different chemistries to label nearby proteins: 1) promiscuous biotin ligases, such as BioID, or 2) peroxidases, such as ascorbate peroxidase (APEX). BioID-based technologies are simple to use and only require biotin and ATP as substrates. In cases where the biological system requires a constant supply of biotin, BioID is continuously active and consumes biotin, resulting in persistent labeling −a major challenge for time-resolved studies and in vivo application[20]. A recently developed light-activatable variant, LOV-turbo[21], is a remarkable improvement, however, comes at the cost of a more complex experimental setup. APEX-based approaches require two substrates: a tyramide, which is oxidized to produce a phenoxyl radical that reacts with nearby targets, and hydrogen peroxide ($H_2O_2$), which is reduced to water[22]. $H_2O_2$ supplementation provides control over the enzymatic activity, yet, endogenous cellular peroxidases can also use $H_2O_2$ to oxidize tyramides[2,23]. To account for such potential non-specific labeling, experimental designs include complex, time- and resource-consuming specificity controls, such as mislocalized APEX transgenes or genetic ablation of the target structure[24]. Most critically, APEX requires $H_2O_2$ to be supplied in high concentrations (mM), which induces oxidative damage in virtually all biological contexts posing a significant challenge for in vivo studies[25–27].

Here, we show that many commonly used cell culture models are incompatible with previous APEX2-based proximity labeling methods. Undesired background often exceeds APEX2-mediated proximity biotinylation due to endogenous peroxidase activities when potentially toxic $H_2O_2$ is added externally. In the work presented here, we could overcome these limitations of APEX-based proximity labeling by employing the enzyme D-amino acid oxidase (DAAO) from *Rhodotorula gracilis*[28] to locally generate $H_2O_2$. Thereby, APEX2-mediated biotinylation is rendered dependent on an enzyme cascade, yielding a more versatile in situ APEX activation (iAPEX) system, which 1) expands the applicability to additional biological systems, 2) reduces toxicity by avoiding addition of exogenous $H_2O_2$, and 3) increases specificity of APEX labeling to circumvent complex genetic controls. Using this methodology, we could successfully determine the proteomes of primary cilia of cell types *hitherto* inaccessible to conventional APEX proximity labeling. In addition, we show that iAPEX is a versatile method enabling effective and organelle-specific protein labeling with superb spatial resolution on lipid droplets (LDs) and mitochondria, which provides the potential to probe dynamic protein interactions at membrane-contact sites with sub-organelle resolution. Finally, we provide proof-of-concept experiments for the in vivo application of iAPEX in *Xenopus laevis*.

## Results

### D-amino acid oxidase can activate ascorbate peroxidase

Quantitative proteomics on subcellular microdomains is technically challenging. Since proteomic information on primary cilia is limited

to few cell types[13,18], we aimed to determine cilia proteomes of cell lines commonly used to study primary cilia by employing cilia-APEX2, an experimental setup we have successfully applied to study the ciliary proteome in a quantitative and time-resolved manner in kidney epithelial cells[17]. As APEX2-based proximity labeling is widespread, we envisioned an easy transfer of the methodology to other cell types. Yet, performing APEX2 labeling reactions using hydrogen peroxide ($H_2O_2$) resulted in various degrees of background biotinylation within cell types of interest, such as C2C12 myoblasts, 3T3-L1 pre-adipocytes, and NIH/3T3 fibroblasts (Fig. 1a). While in IMCD3 cells biotinylation was specific to the expression and localization of the APEX2 enzyme (visualized by GFP fluorescence of NPHP3[1–200]-GFP-APEX2), biotinylation was observed throughout the cell bodies of C2C12, 3T3-L1 and NIH/3T3 cells, with visibly higher overall signal than in cilia-APEX2 expressing IMCD3 cells (Fig. 1a). After generating an NIH/3T3 cell line that stably expresses cilia-APEX2, we performed APEX2 labeling reactions by $H_2O_2$ addition and investigated the amounts of biotinylation by SDS-PAGE and western blotting (Fig. 1b). Surprisingly, biotinylation in NIH/3T3 cells was independent of the presence of the cilia-APEX2 enzyme and greatly surpassed the amounts observed in the well-established IMCD3 cell line (Fig. 1b, lanes 4 *vs.* 6), which indicated excessive background biotinylation by endogenous peroxidases.

To overcome non-specific proximity labeling and avoid external addition of $H_2O_2$, we expressed D-amino acid oxidase (DAAO) from *Rhodotorula gracilis* that oxidizes D-amino acids, the rare enantiomers of the predominant L-amino acids, to produce $H_2O_2$ intracellularly[28–31]. We specifically decided for this well characterized and established DAAO enzyme as it has been used successfully in several in vivo applications in human and rodent tissue[30,32]. To specify and restrict the subcellular localization of $H_2O_2$ production, we targeted DAAO to primary cilia by fusing it to the first 200 amino acids of the ciliary protein NPHP3, which we term cilia-DAAO (Fig. 1c). We hypothesized that locally produced $H_2O_2$ would be immediately used by nearby APEX2 to oxidize biotin tyramide (BT) for proximity labeling (Fig. 1d). To confirm the functionality of this enzymatic cascade in primary cilia, we generated an IMCD3 cell line stably expressing both cilia-APEX2 and cilia-DAAO, which localize specifically to primary cilia (Fig. 1e). In this cell line, proximity biotinylation in primary cilia could be achieved in the presence of biotin tyramide either by addition of $H_2O_2$ (to activate APEX2 directly) or by providing the DAAO substrate D-alanine (D-Ala) (Fig. 1e). Although small molecules can diffuse freely between the cilium and the cytoplasm[33], the functionality of the cascade required DAAO to be localized to cilia, as a DAAO enzyme localized to the cytosol (cyto-DAAO) did not activate cilia-APEX2 (Supplementary Fig. 1a). This suggests that $H_2O_2$ produced by DAAO in the cytoplasm does not diffuse into primary cilia, probably due to rapid detoxification. Interestingly, in cilia with strong biotin signals we observed a reduction in the cilia-DAAO signal (Supplementary Fig. 1b,c). As cilia-DAAO is detected via the FLAG epitope, we interpret this anti-correlation as a potential biotinylation of the tyrosine residue within the FLAG epitope, which may mask antibody binding.

D-amino acids are inert in most biological systems but show biological activity in rare instances, such as D-serine as a putative gliotransmitter[34–36]. Therefore, we tested different amino acids and derivatives as potential DAAO substrates for APEX-based proximity labeling in the cilia-APEX2 and cilia-DAAO expressing IMCD3 cell line. Except for D-valine, all D-amino acids tested induced biotin tyramide-dependent biotinylation in primary cilia in IMCD3 cells, which confirmed suitability and stereo-specificity of these substrates for DAAO-dependent APEX labeling (Supplementary Fig. 1d). Taken together, our experiments confirmed the functionality of the DAAO-APEX enzymatic cascade, which we term "in situ APEX activation" ("iAPEX") proximity labeling.

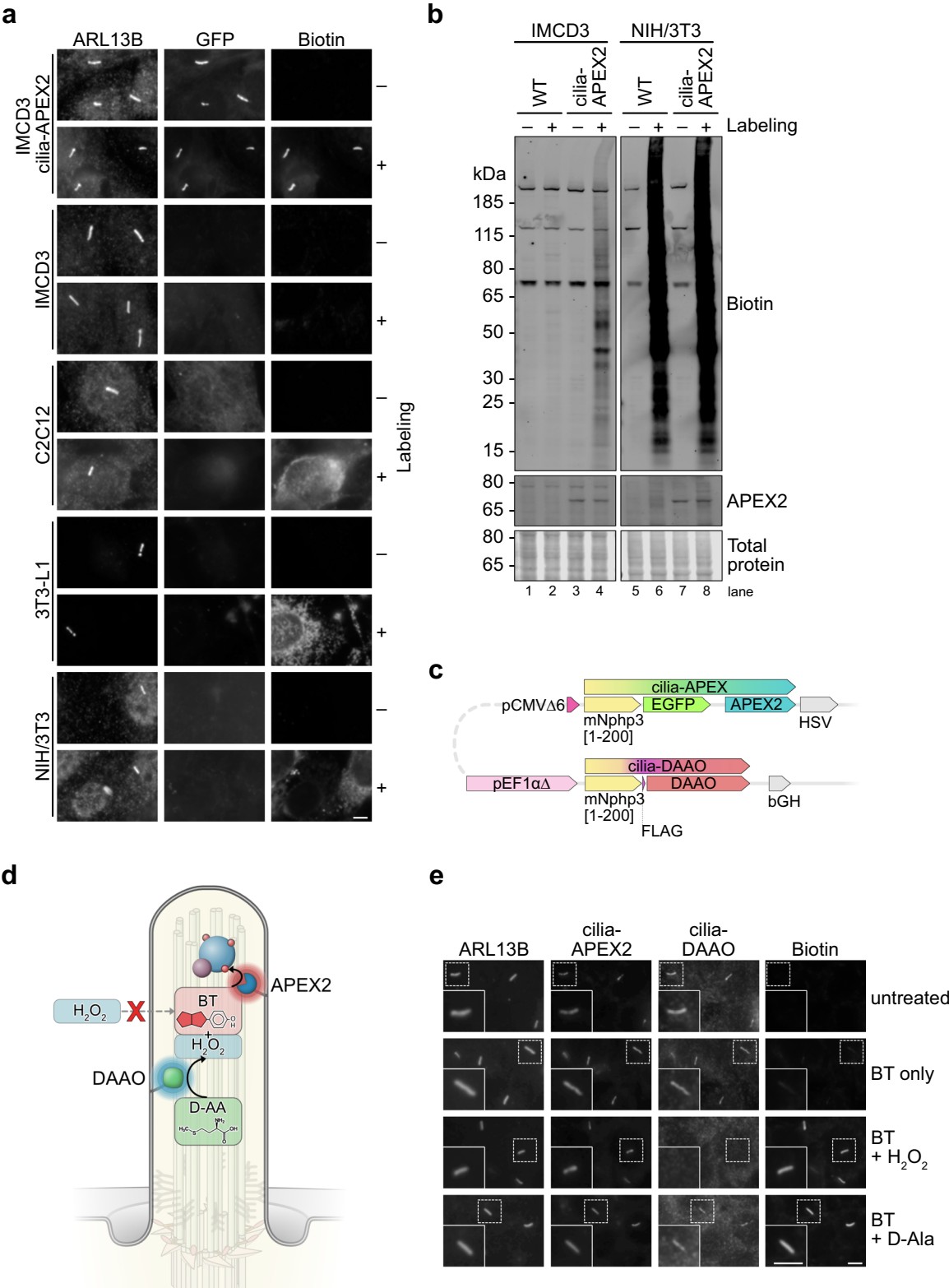

## Local hydrogen peroxide production minimizes toxicity

As the local restriction of APEX2 activation within primary cilia may limit its application for whole-cilium proteomics, we assessed the sub-ciliary localization of biotinylated proteins by ultra-structure expansion microscopy (U-ExM)[37]. U-ExM confirmed that both cilia-APEX2 and cilia-DAAO were confined to the membrane of the primary cilium (Fig. 2a and Supplementary Fig. 2a)

with varying degrees of co-localization. However, after activation of the iAPEX cascade, biotinylation was not restricted to the membrane and could be detected throughout the entire cilium, indistinguishable from the activation by external addition of $H_2O_2$ (Fig. 2b and Supplementary Fig. 2b), indicating that iAPEX labeling generates sufficient phenoxyl radicals to probe the entire cilium.

**Fig. 1 | Background biotinylation in various cell types limits APEX-based proximity labeling applications. a** Immunofluorescence micrographs of IMCD3, C2C12, 3T3-L1, NIH/3T3, and IMCD3 cells stably expressing NPHP3[1–200]-GFP-APEX2 (cilia-APEX2). Cilium formation was induced by 24 h growth factor deprivation. Cells were left untreated (−) or subjected to APEX2 proximity labeling (+) by incubation with 500 μM biotin tyramide (BT) for 30 min followed by 1 mM hydrogen peroxide ($H_2O_2$) for 1 min (3T3-L1) or 3 min (remaining cells). After fixation, primary cilia were visualized by anti-ARL13B antibody staining, cilia-APEX2 by GFP fluorescence, and biotin by fluorescently labeled streptavidin. Same imaging parameters for all images. **b** Wild-type (WT), cilia-APEX2-expressing IMCD3 and NIH/3T3 cells were lysed before (−) or after (+) APEX2 proximity labeling, followed by SDS-PAGE and western blot analysis (*n* = 4 independent experiments). Biotin detection as in (**a**), equal protein loading confirmed by total protein stain. **c** Diagram of cilia-iAPEX expression cassette with NPHP3[1–200]-GFP-APEX2 (cilia-APEX2) and NPHP3[1–200]-FLAG-DAAO (cilia-DAAO) transgenes in head-to-head orientation. A vector containing this cassette allows stable genomic integration via Flp-In recombinase and low-level expression of cilia-APEX2 and cilia-DAAO from truncated cytomegalovirus promoter (pCMVΔ6) and EF1α promoter lacking the TATA box (pEF1αΔ), respectively[107]. Enzymes are fused to N-terminal cilia-targeting sequences (amino acids 1-200 of murine Nephrocystin-3 (mNphp3)) and tagged with enhanced GFP (EGFP) for APEX2 and FLAG for DAAO. **d** Schematic: primary cilium harboring the in situ APEX activation (iAPEX) proximity labeling enzymes. APEX2 and DAAO are genetically targeted to primary cilia using constructs displayed in (**c**). A D-amino acid (D-AA) serves as DAAO substrate for in situ hydrogen peroxide ($H_2O_2$) production through oxidative deamination. Locally produced $H_2O_2$ and biotin tyramide (BT) are APEX2 substrates for proximity biotinylation, overcoming the need for external $H_2O_2$ addition (red X). **e** Immunofluorescence micrographs showing APEX2 proximity labeling in primary cilia of IMCD3 cells stably expressing the cilia-targeted iAPEX enzyme cascade (*n* = 5 independent experiments). APEX2 proximity labeling: incubation with biotin tyramide (BT) for 30 min and $H_2O_2$ for 3 min. DAAO-facilitated proximity labeling: D-alanine (D-Ala, 10 mM) added during BT incubation for 30 min. Cilia-DAAO is detected by anti-FLAG antibodies, all others as in (**a**). All scale bars = 5 μm.

To identify the minimum concentrations of D-amino acids required for efficient labeling, we titrated the DAAO substrates D-alanine (D-Ala) and D-methionine (D-Met) and assessed cilia-iAPEX-catalyzed biotinylation efficiency by immunofluorescence microscopy (Fig. 2c and Supplementary Fig. 2c). Quantitation of biotinylation in primary cilia showed a concentration-dependent increase in biotinylation for both D-Ala and D-Met (Fig. 2c). High concentrations of both D-amino acids led to stronger biotinylation in cilia compared to 3 min labeling with $H_2O_2$, although D-Ala did not reach the same levels as D-Met. For D-Met saturating signals were achieved at 4 mM (Fig. 2c), while notable biotin signals could be observed at concentrations as low as 125 μM when DAAO-catalyzed proximity biotinylation was performed for 30 min (Fig. 2d). We therefore focused on the use of D-Met as DAAO substrate for our applications. As biotin tyramide exhibits moderate cell permeability[2], we aimed to increase the temporal resolution of biotinylation by pre-incubating cells with biotin tyramide before D-Met addition. A time course of the labeling reaction revealed that 5 min incubation with 10 mM D-Met after 30 min biotin tyramide pre-incubation was comparable to short activation (3 min) with 1 mM $H_2O_2$ (Fig. 2e). While we incubated for 2 min with $H_2O_2$ in a former study[17], 5 min labeling is still compatible with our previous temporal resolution. Further experiments revealed that 5 min pre-incubation with biotin tyramide was sufficient, as longer pre-incubation times did not increase the labeling efficiency (Supplementary Fig. 2d). Interestingly, pre-incubation with D-Met to initiate local $H_2O_2$ production prior to biotin tyramide addition decreased biotinylation in a time-dependent manner (Supplementary Fig. 2d), indicating that prolonged $H_2O_2$ production might interfere with APEX2 function. Thus, the overall temporal resolution that can be achieved in IMCD3 cells is 5 min, which compares favorably to recently developed proximity labeling methods[21,38].

To assess the consequence of local $H_2O_2$ production and biotinylation on the function of primary cilia, we investigated intraflagellar transport in IMCD3 cells transiently transfected with IFT38-GFP-APEX2 and cilia-localized DAAO by live-cell microscopy (Supplementary Movie 1) and could not observe any obvious difference after 30 min $H_2O_2$ production (Supplementary Movie 2) or iAPEX labeling (Supplementary Movie 3).

Although the basic functions of primary cilia still appeared intact in the timeframe of iAPEX labeling, $H_2O_2$ in primary cilia is likely to cause oxidative damage locally. To assess potential toxicity of the iAPEX system for the entire cell, we determined the $H_2O_2$ production by cilia-DAAO employing oxygen ($O_2$) consumption measurements as DAAO activity converts $O_2$ to equimolar amounts of $H_2O_2$[39]. When blocking cellular respiration with oligomycin the cilia-iAPEX IMCD3 cell line consumed approximately 1 fmol/(min×cell) $O_2$ at steady state (Fig. 2f). After addition of D-Met the $O_2$ consumption increased to about 1.2 fmol/(min×cell) (Fig. 2f). This rise in $O_2$ consumption was D-amino-acid- and cilia-DAAO-dependent, as it was not observed in the parental cell line (Fig. 2g). Although D-Ala led to a comparable maximum $O_2$ consumption rate as D-Met, our kinetic analysis indicated a 30 min delay to reach this maximum (Fig. 2h), which agrees with the observed differences in iAPEX-catalyzed biotinylation (see Fig. 2c). Assuming an average cell volume of 4.000 fL, the increase in $O_2$ consumption of 0.2 fmol/(min×cell) would result in sub-μM $H_2O_2$ concentrations within 1 sec of DAAO activity if the cell completely lacked mechanisms to detoxify $H_2O_2$. However, as physiological redox signaling requires an existing, potent antioxidant system[40], our data indicate that the amount of $H_2O_2$ produced by cilia-DAAO is within physiological $H_2O_2$ concentrations and can therefore be considered non-toxic for most cell types.

## Cilia-iAPEX locally restricts proximity biotinylation and prevents off-target biotinylation

To test whether the iAPEX labeling cascade overcomes the high background observed in select cell lines (see Fig. 1a, b), we introduced cilia-iAPEX into NIH/3T3 cells using the Flp-In system, and isolated a clonal cell line, in which both enzymes localized to primary cilia (Fig. 3a). Most importantly, iAPEX labeling was specific to primary cilia in this cell line and indicated that spatially restricted $H_2O_2$ production by cilia-DAAO prevented non-specific biotinylation of other cellular structures, as observed after direct activation of cellular peroxidases (Fig. 3a). SDS-PAGE and western blot analysis of whole-cell lysates from cilia-iAPEX expressing NIH/3T3 cells confirmed high background biotinylation when using $H_2O_2$ as a substrate, while DAAO activation resulted in markedly reduced but specific biotinylation (Fig. 3b, lanes 11 *vs.* 12). Interestingly, even in whole-cell lysates from the established IMCD3 cell line we observed overall stronger signals when using $H_2O_2$ compared to D-Met-based proximity labeling (Fig. 3b, lanes 8 *vs.* 9), despite weaker biotin signals in primary cilia (see Fig. 2c). To gain a deeper understanding of the events occurring during APEX labeling, we established a live-cell imaging setup to visualize the subcellular localization of peroxidase activity by the oxidation of the peroxidase substrate Amplex UltraRed (AmUR) to a fluorescent resorufin product[41,42]. After loading cells that stably express cilia-iAPEX with AmUR, we noticed a burst in peroxidase activity throughout the entire cell shortly after $H_2O_2$ addition, which ceased over time when only the APEX activity in the primary cilium remained (Fig. 3c and Supplementary Movie 4). In contrast, local production of $H_2O_2$ by cilia-DAAO prevented non-specific peroxidase activity, as we observed resorufin signals exclusively in primary cilia for prolonged labeling times (Fig. 3d and Supplementary Movie 5). These results indicate that other cellular peroxidases are capable of oxidizing substrates, such as biotin

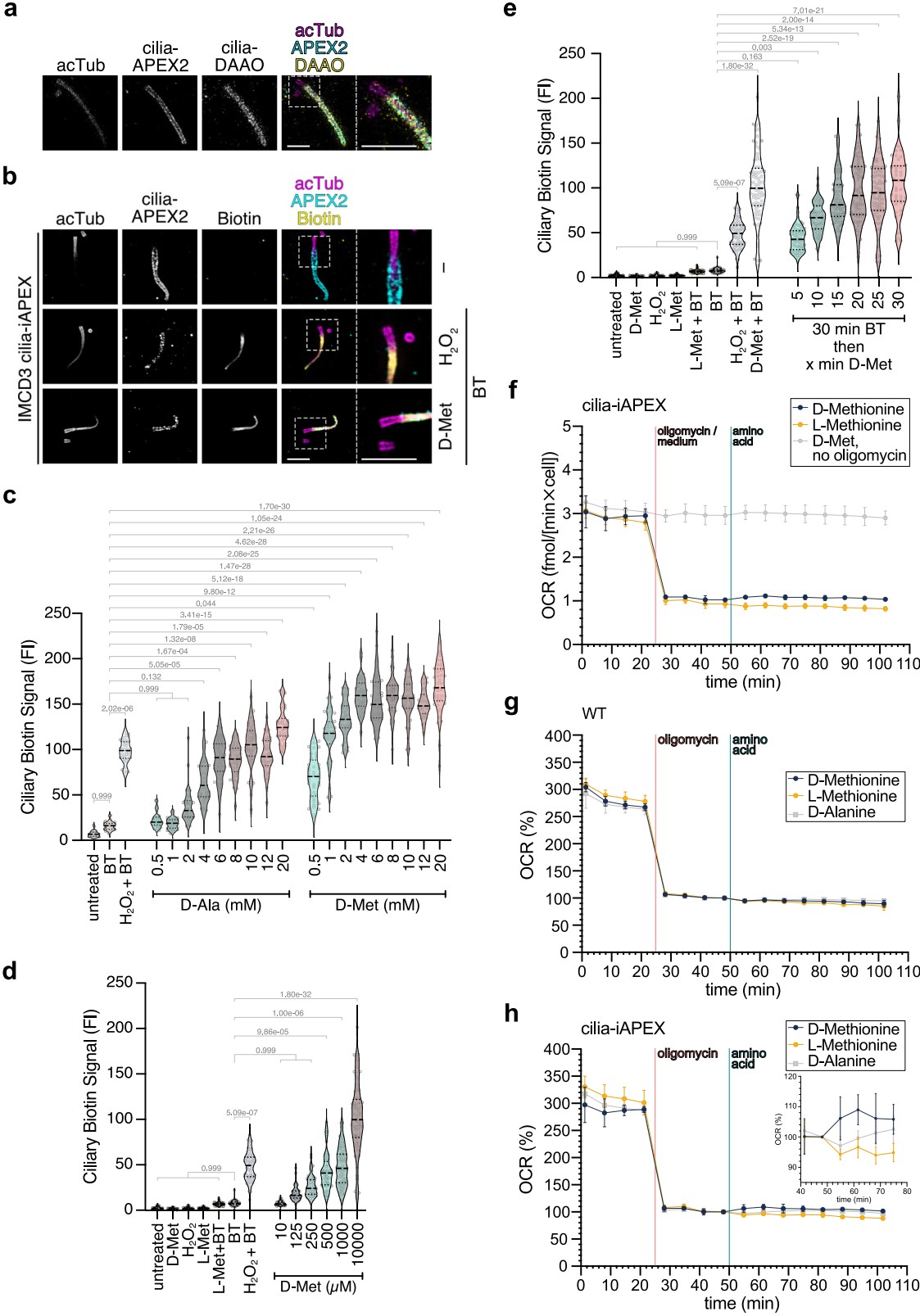

tyramide to biotinylate nearby proteins by proximity labeling when $H_2O_2$ is added to the cells. We further hypothesize that the observed initial burst in peroxidase activity may cause significant non-specific biotinylation and thereby high background when studying proteomic environments of targets that are expressed at low levels, such as for primary cilia proteomics.

## Cilia-iAPEX proteomics increases specificity and sensitivity of cilia protein identification

To directly compare the iAPEX- with the APEX2-based proximity labeling method as a discovery tool in proteomics applications, we performed iAPEX (DAAO-dependent) and APEX2 ($H_2O_2$-induced) proximity labeling in cilia-iAPEX IMCD3 cells using desthiobiotin

**Fig. 2 | In situ D-amino acid oxidase-mediated hydrogen peroxide production enables APEX2 proximity labeling. a, b** Ultrastructure expansion microscopy (U-ExM) confocal images showing cilia-targeted iAPEX localization and proximity labeling. Cells were fixed, cross-linked, and embedded in a water-expandable gel. **a** Expanded primary cilium from an RPE-1 cell line stably expressing cilia-iAPEX reveals membrane localization of APEX2 (cyan) and DAAO (yellow), probed with antibodies against GFP and ALFA tag, respectively. Acetylated tubulin (acTub) is shown in magenta. **b** U-ExM micrographs of IMCD3 cells stably expressing cilia-iAPEX demonstrates biotinylation within the entire cilium. Cells were either untreated (−), labeled with biotin tyramide (BT) and $H_2O_2$, or 10 mM D-methionine (D-Met). Biotin visualization as in Fig. 1 (yellow). Scale bars = 5 µm (adjusted to expansion factors = 4 (**a**) and 4.3 (**b**)). **c–e** Quantification of absolute ciliary biotin signals in micrographs obtained from proximity labeling experiments performed in IMCD3 cilia-iAPEX cell line shown as violin plots. Dotted and dashed lines represent quartiles and medians, respectively. **c** Type and concentration of D-amino acid affect biotinylation. Indicated concentrations of D-Ala or D-Met were incubated for 30 min. $n = 40$ cilia per condition. **d** DAAO shows stereoselectivity for D-amino acids and allows labeling with low concentrations of D-Met. $n = 77$ cilia from two experiments. **e** Shorter substrate incubation leads to comparable biotinylation as $H_2O_2$-induced labeling. Incubation with BT and 10 mM D-Met for indicated times. $n = 20$ cilia from two experiments. Where indicated, 1 mM $H_2O_2$ was incubated for 3 min. Data analysis: one-sided Kruskal-Wallis test followed by Dunn's multiple comparison. Numbers indicate $p$ values. **f–h** D-Met-activated cilia-DAAO generates minute amounts of $H_2O_2$. $O_2$ consumption rates (OCR) measured by Seahorse metabolic flux analysis with an average of 30.000 cells (coefficient of variation <5.0%). **f** OCRs of cilia-iAPEX IMCD3 cells determined after treatment with or without oligomycin (blocking cellular respiration), followed by D-Met or L-Met addition to activate DAAO. **g** OCRs of wild-type (WT) and **h** cilia-iAPEX IMCD3 cells were recorded and normalized to OCR after oligomycin treatment before addition of indicated amino acids (100%). Insert shows zoom. Lines depict means, error bars standard deviations ($n = 3$). Source data are provided as a Source Data file.

tyramide (DTBT) as a substrate, as this allowed competitive elution of APEX-biotinylated proteins after isolation by streptavidin chromatography (Fig. 4a). Abundant non-specific biotinylation in APEX2-based proximity labeling setups requires controls to precisely assess the background[43,44]. To this end, for cilia proteomics we previously expressed cilia-APEX2 in $Cep164^{-/-}$ cells that lack primary cilia[17]. Triplicates of the iAPEX-labeled samples and duplicates of the controls were analyzed by SDS-PAGE and western blotting. Our analyses confirmed reduced biotinylation by iAPEX compared to APEX2 labeling (Supplementary Fig. 3a, lanes 6-7 vs. 8-10), while several cilia components, represented by IFT88 and IFT57, were isolated more efficiently after iAPEX labeling (Fig. 4b, lanes 13-15 vs. 11,12). This indicated a higher sensitivity of cilia-iAPEX compared to previous setups, while the $Cep164^{-/-}$ controls confirmed specificity of isolation, as no ciliary proteins were isolated in the absence of cilia (Fig. 4b, lanes 9,10). To quantitatively assess the performance of cilia-iAPEX vs. cilia-APEX2, isolated proteins were digested by trypsin, labeled with tandem-mass-tags (TMT) and analyzed by liquid chromatography and multistage mass spectrometry (LC-MS[3])[5,17,45,46] (Fig. 4a). We quantified the relative abundances of 5982 identified proteins within the individual samples (see Supplementary Data 1). Replicates showed high similarity across conditions, as evidenced by pairwise multiscatter plots and high Pearson correlation coefficients (Supplementary Fig. 3b). When assessing candidate ciliary proteins by statistical analysis of relative enrichments between cilia-iAPEX and control samples, we applied stringent TMT enrichment ratios of $2^3$ which resulted in 175 high confidence candidate cilia proteins (Supplementary Fig. 3c). Surprisingly, within the same experiment the same TMT enrichment ratio cutoff between cilia-APEX2 and control samples identified 799 putative cilia proteins (Supplementary Fig. 3d). A direct comparison showed that the enrichment of known cilia proteins was similar in both approaches, however, cilia-iAPEX proteomics separated known cilia proteins much better from non-cilia proteins (Fig. 4c). Gene Ontology (GO) term enrichment analyses confirmed higher specificity of the iAPEX setup, as evidenced by the absence of non-ciliary processes and the lower $p$ values of ciliary categories (Supplementary Fig. 3e,f). Hierarchical clustering of the relative protein abundances within the experiment confirmed high reproducibility of the replicate samples (Supplementary Fig. 4a). Proteins enriched in both iAPEX and APEX2-labeled samples formed three clusters highly enriched in known cilia proteins (Fig. 4d, Supplementary Fig. 4b and Supplementary Data 1), which covered 38% of the cilia-APEX2 proteome that was based on three independent datasets[17]. Interestingly, the proteins in these cilia clusters contributed to the lower correlation with control samples, underscoring their specificity and higher abundance in the iAPEX samples (see Supplementary Fig. 3b, orange). A GO term enrichment analysis revealed high statistical significance of components associated with cilia and related microtubule-based structures (Fig. 4e),

while our previous cilia-APEX2 proteome contained many non-ciliary categories[17], suggesting false-positive hits. Agreeingly, our cluster analysis also identified proteins that were enriched only in the $H_2O_2$-treated samples, in wild-type and $Cep164^{-/-}$ cells, that formed four clusters (Fig. 4f and Supplementary Fig. 4c). GO term enrichment analysis of these clusters identified a large fraction of proteins located in the endoplasmic reticulum (ER) (Fig. 4g), suggesting that ER resident peroxidases in IMCD3 cells can biotinylate nearby proteins in a $H_2O_2$-dependent manner. Such peroxidases may cause non-specific labeling and contribute to potential false-positive hits in conventional APEX2 labeling setups. To rule out that the proteins in the false-positive clusters were absent from the iAPEX labeled samples for technical reasons, due to an oxidation-dependent mass shift, we investigated the oxidized peptides in our dataset. When comparing peptide distributions of oxidized and non-oxidized peptides across all samples, we observed a highly similar distribution in the false-positive clusters and among peptides from IFT proteins (Supplementary Fig. 4d), supporting the notion that the proteins identified by conventional APEX2 labeling are likely false-positives. Taken together, iAPEX-based proteomics shows high sensitivity to analyze the proteome of subcellular microdomains and significantly reduces the number of false-positives by lowering background biotinylation activities.

### Determining the cilia-iAPEX proteome of NIH/3T3 cells

As iAPEX proximity labeling allowed the specific biotinylation of proteins in NIH/3T3 cells (see Fig. 3), we sought to gain proteomic information on the ill-defined proteome of NIH/3T3 primary cilia. By combining hierarchical clustering with the increased specificity of the iAPEX system, we envisioned that the cilia proteomes could be investigated without the need for genetic background controls, such as the $Cep164^{-/-}$. Therefore, we utilized the cilia-iAPEX NIH/3T3 cell line and included two simple specificity controls that prevent iAPEX labeling: 1) omitting the APEX2 substrate desthiobiotin tyramide, and 2) replacing the DAAO substrate D-Met for L-Met, which does not activate DAAO (Fig. 5a, see also Fig. 2d, e). Replicate samples were prepared, followed by streptavidin chromatography and analysis by SDS-PAGE and western blotting. The analysis revealed biotinylation of proteins in the presence of desthiobiotin tyramide (Fig. 5b), independent of DAAO activation, which suggests background labeling mediated by endogenous $H_2O_2$. Nonetheless, we observed an increase in biotinylation when DAAO was activated by D-Met (lanes 5-7 vs. 11-13). Importantly, there was a clear enrichment of the cilia proteins IFT57 and IFT88 only when iAPEX labeling was performed (Fig. 5c), which indicated that cilia proteins were only labeled when both enzymes were activated and omitting either substrate could serve as specificity controls. We therefore continued with TMT labeling and LC-MS[3] analysis and quantified the relative abundances of the 6067 quantified proteins

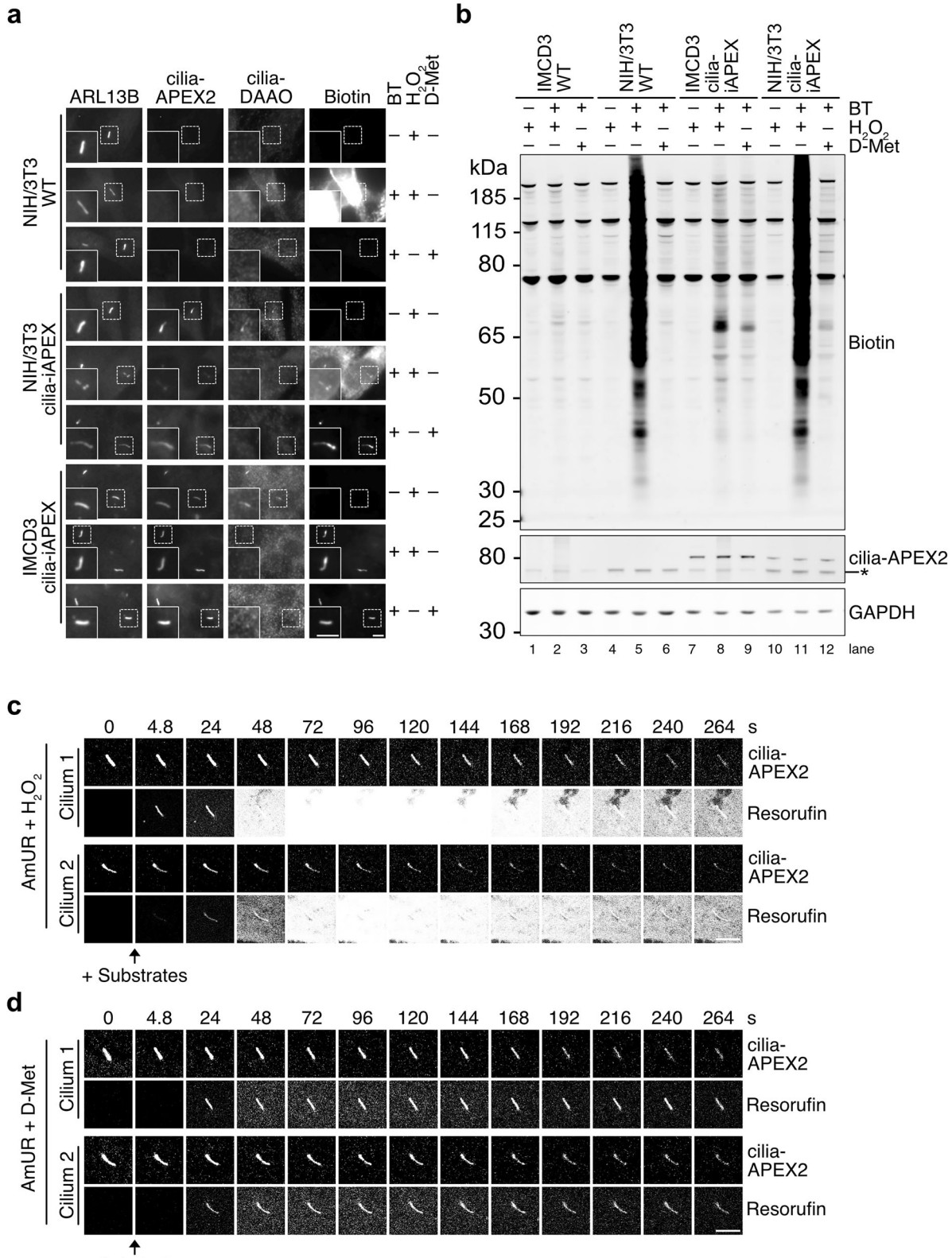

(Supplementary Data 2). After hierarchical clustering most clusters revealed similarly distributed protein abundances (Supplementary Fig. 5a), with high similarity across replicates and conditions as revealed by high Pearson correlation coefficients in pairwise multi-scatter plots (Supplementary Fig. 5b). In 14 clusters, proteins were enriched in the six desthiobiotin tyramide-treated samples, indicating biotinylation (see also Fig. 5b). Only four out of those 14 clusters

contained proteins highly enriched in the iAPEX-labeled samples and included many known cilia proteins, such as IFT subunits, BBSome components and molecular motors (Fig. 5d, Supplementary Fig. 5c and Supplementary Data 2). These clusters also contained several proteins previously not implicated in cilia biology. When comparing the cilia-iAPEX proteomes of IMCD3 cells with NIH/3T3 cells, we noticed an overlap of below 50% (Fig. 5e and Supplementary Data 3), which could

**Fig. 3 | iAPEX enables cilia-specific biotinylation in NIH/3T3 bypassing high cellular background. a** Immunofluorescence micrographs of NIH/3T3 WT and cilia-iAPEX cells with IMCD3 cilia-iAPEX cells as control ($n = 5$ independent experiments). Cells were left untreated, labeled using BT and $H_2O_2$, or BT and D-Met, as indicated. **b** Western blot analysis of WT and cilia-iAPEX-expressing IMCD3 and NIH/3T3 cells. Cells were lysed before or after BT and $H_2O_2$, or BT and D-Met treatment ($n = 4$ independent experiments). Asterisk marks cross-reactive band of the anti-GFP antibody. Please note that the GAPDH loading control was not analyzed on the same gel, but from identical samples analyzed in parallel. **c, d** Live-cell confocal imaging micrographs were captured to observe peroxidase-dependent Amplex UltraRed (AmUR) oxidation to resorufin in IMCD3 cells stably expressing cilia-APEX2. **c** Cells were treated with 50 μM AmUR together with 1 mM $H_2O_2$ where indicated. Resorufin and GFP fluorescence of cilia-APEX2 were monitored at 4.8-second intervals over a total duration of 264 s (see also Supplementary Movie 4). **d** AmUR oxidation reveals exclusive cilia-APEX2 activity when DAAO-dependent $H_2O_2$ production was performed after addition of 50 μM AmUR and 10 mM D-Met (see also Supplementary Movie 5). Two cilia are shown per condition. Scale bars = 5 μm in all panels.

suggest cell type-specific heterogeneity of the cilia proteome. While there was a significant overlap between the proteomes of our cilia-iAPEX proteome with our previous cilia-APEX2 study[17], only few core cilia proteins were common with a TurboID-based cilia proteome in NIH/3T3 cells[18] (Supplementary Fig. 6a), which may be due to different proximity labeling strategies. Our datasets, however, covered twice as many ciliopathy-associated proteins, as defined by the CiliaMiner[47] (Supplementary Fig. 6b). A direct comparison between the cilia-APEX-based studies in IMCD3 cells showed that we identified 16 more ciliopathy genes in our previous, more extensive cilia-APEX2 work[17], most of which were membrane and transition zone proteins (Supplementary Fig. 6c). While 41 ciliopathy genes were identified in both studies, here, we found additional 4 microtubule-associated proteins (MAPs) implicated in ciliopathies, suggesting a higher sensitivity for axonemal proteins. While the absolute number of commonly identified proteins in the cilia-APEX2 datasets was bigger, the fraction of commonly identified proteins clearly improved with the cilia-iAPEX method (Supplementary Fig. 6d), likely due to higher specificity (see Fig. 4). Importantly, our datasets from IMCD3 and NIH/3T3 cells contained several proteins previously not implicated in cilia biology, some of which have been associated with human disease. We therefore set out to confirm cilia localization of select candidates by independent methods. To this end we tagged the proteins of interest with an ALFA affinity tag[48], and created IMCD3 cell lines stably expressing the ALFA-tagged proteins of interest and assessed their localization by fluorescence microscopy (Fig. 5f). Indeed, we could confirm the localization of the proteins PSKH1, CUEDC1, and CKAP2L to the primary cilium shaft. We could also validate cilia localization of two so-far uncharacterized mouse homologs of the human open reading frames C19orf44 and C7orf57 in IMCD3 cells, which we termed FCAP71 and FCAP33 (Found in cilia-iAPEX proteome of 71 and 33 kDa), respectively. FCAP33 showed enrichment in primary cilia, whereas FCAP71 localized to the base of cilia, marked by CEP164 (Fig. 5f). We further tested proteins exclusively identified in NIH/3T3 cilia. We transiently transfected the tagged proteins CKAP5 and the formin FHDC1, which showed clear cilia localization in NIH/3T3 cells (Fig. 5g). We noticed that FHDC1 expression led to an increase in cilium length, as previously reported[49]. While the precise functions of these cilia proteins remain to be investigated, we conclude that iAPEX proximity labeling is a powerful, unbiased discovery tool for subcellular proteomics of technically demanding cell lines.

## Spatial restriction of iAPEX labeling and non-cilia applications
Cilia-iAPEX labeling at high D-amino acid concentrations was not restricted to the membrane-localized enzymes as the entire cilium appeared biotinylated (see Fig. 2b). To test whether the iAPEX system could also be used for a more confined labeling, we fused the APEX2 enzyme to GLI2, a transcription factor in the Hedgehog signaling pathway that localizes to the tips of primary cilia[50]. In cells co-expressing GFP-APEX2-GLI2 with cilia-DAAO, biotinylation increased in a D-Met concentration-dependent manner (Fig. 6a and Supplementary Fig. 7a). While higher concentrations of D-Met led to biotinylation across the entire cilium (Fig. 6a, arrows), labeling was restricted to the tip when low concentrations of D-Met were used, indicating that the

spatial resolution of iAPEX labeling can be increased by lowering the amounts of $H_2O_2$ production.

To test whether the iAPEX system enables the analysis of other subcellular organelles with inherent complexities, we directed DAAO and APEX2 across two intracellular membranes into the mitochondrial matrix. We co-transfected cells with plasmids co-expressing mito-APEX2 and soluble GFP to confirm expression, and mito-DAAO-FLAG[51] (Fig. 6b). We could only detect biotinylation in mitochondria above background when biotin tyramide and D-Ala were supplied together, which highlights the versatility of the iAPEX approach.

The membranes of mitochondria as well as the primary cilium limit the diffusion of phenoxyl radicals produced by APEX2 to the membrane-bound structures, however, most intracellular organelles are directly exposed to the cytosol. Therefore, we asked whether the iAPEX cascade could be applied in other cellular contexts. Lipid droplets (LDs) are cytosolic lipid storage organelles that are surrounded by a phospholipid monolayer[52] (Fig. 6c). While many LD proteins, such as UBXD8[53–55] co-exist on LDs and the endoplasmic reticulum (ER), other proteins localize exclusively to LDs (e.g. PLIN5[56,57]). To achieve spatially restricted biotinylation by the UBXD8 pool on LDs, we designed a dual-targeting approach: while APEX2 was fused to UBXD8, which resides on both LDs and the ER, DAAO was fused to PLIN5 to drive APEX2 activation specifically on the LD surface (Fig. 6c). As expected, DAAO-PLIN5 predominantly localized to LDs, while UBXD8-APEX2 localized to LDs and the ER (Fig. 6d), providing a system to test whether iAPEX labeling can be selectively activated on LDs despite APEX2 availability on the ER.

Conventional APEX labeling by activating UBXD8-APEX2 with externally added $H_2O_2$ resulted in biotinylation across the entire cytosol, with only slight accumulation on the LD surface (Fig. 6d). Similarly, 15 min iAPEX labeling using 10 mM D-Met showed biotinylation of the LD surface in addition to significant cytosolic background. To test whether the biotinylation of cytosolic proteins stems from prolonged and excessive generation of $H_2O_2$ by DAAO, we reduced the D-Met concentrations and incubation times, which significantly improved the spatial restriction of biotinylation to the LD surface. When labeling with 0.25 mM D-Met for 2 min, biotin signals were predominantly observed on the LD surface (Fig. 6d), with minor background signals of endogenous mitochondrial biotin that was also detected in absence of biotin tyramide. The selective activation of LD-localized, but not ER-localized, UBXD8-APEX2 by LD-targeted DAAO-PLIN5 demonstrates the spatial precision of the iAPEX system (Fig. 6c). It further underscores iAPEX's effectiveness and versatility, including its potential to probe dynamic protein interactions at membrane-contact sites with sub-organelle resolution.

## Expanding the iAPEX enzyme cascade to other cell types and organisms for in vivo applications
For cilia proteomics in IMCD3 and NIH/3T3 cells, the iAPEX transgenes were integrated into existing FRT sites on the genomes of the respective cell lines. For a more versatile delivery of the transgenes into additional cell lines of interest, such as C2C12 myoblasts, 3T3-L1 pre-adipocytes or primary cells, we engineered two plasmids for packaging the cilia-iAPEX transgenes into lentiviral particles for cell infection (Fig. 7a). One

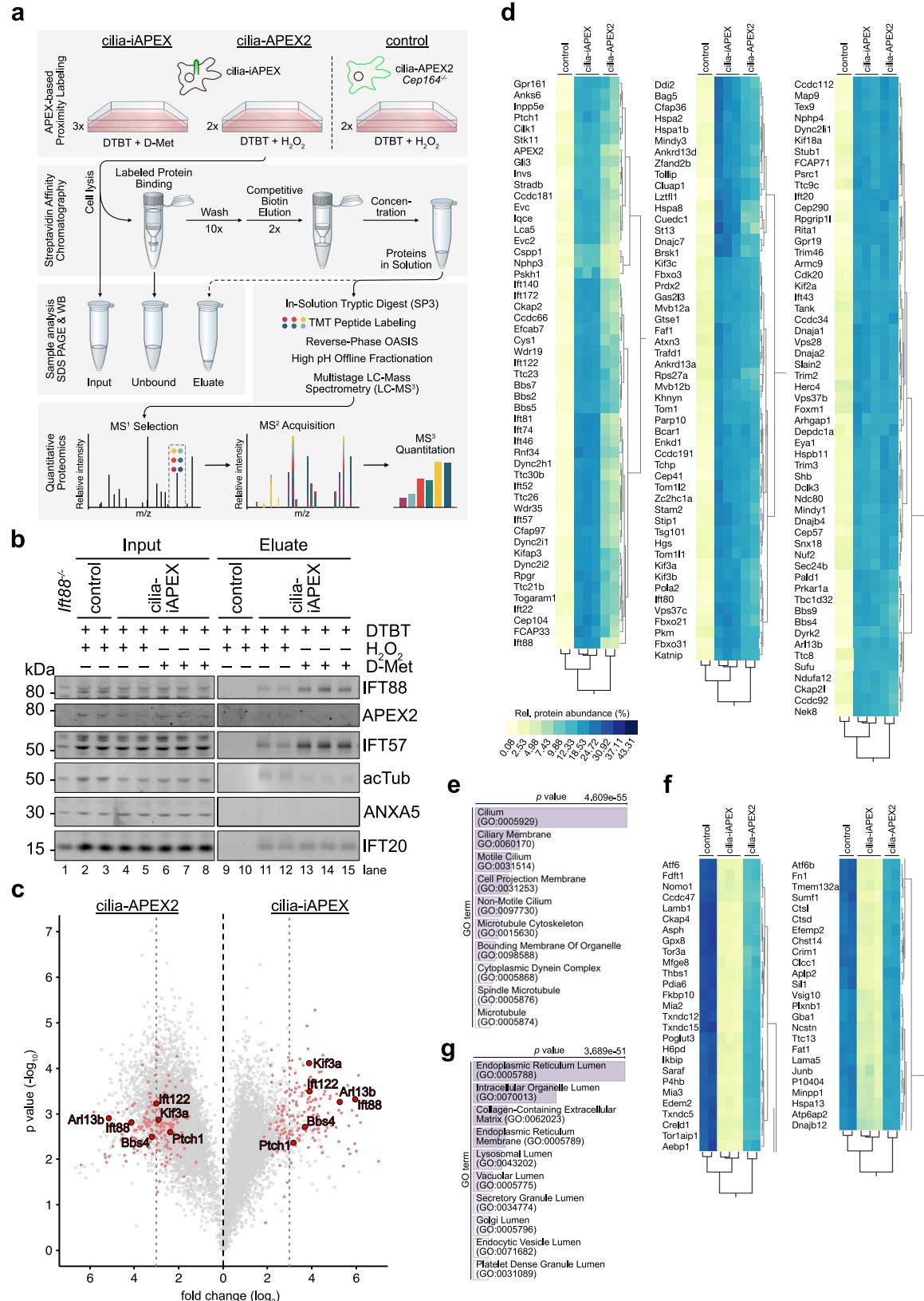

construct expresses NPHP3$^{1–200}$-GFP-APEX2 and NPHP3$^{1–200}$-FLAG-DAAO-ALFA separated by an internal ribosome entry site (IRES), which resembles the cilia-iAPEX used in the Flp-In system (see Fig. 1c). A second construct expresses NPHP3$^{1–200}$-GFP-APEX2 and PKHD1$^{CTS}$-FLAG-DAAO-ALFA separated by a T2A peptide. The T2A peptide required the use of amino acids 3868-3886 of fibrocystin (PKHD1$^{CTS}$) for cilia targeting[58], such that it creates a smaller construct in which both

enzymes are transported to cilia by independent mechanisms. After lentivirus infection with the cilia-iAPEX construct, GFP-positive cells were sorted by fluorescence-activated cell sorting (FACS) and subjected to $H_2O_2$-based APEX2 labeling or iAPEX labeling using D-Met. Analysis by fluorescence microscopy revealed that both cilia-APEX2 and cilia-DAAO exhibited specific localization to primary cilia in 3T3-L1, C2C12, IMCD3 and RPE-1 cells (Fig. 7b and Supplementary Fig. 7b). Despite the

**Fig. 4 | Quantitative primary cilia proteomics using iAPEX outperforms conventional APEX2-based system in IMCD3 cells. a** Schematic: cilia-iAPEX-based proximity labeling workflow for proteomic analysis of IMCD3 primary cilia. iAPEX labeling in cilia-iAPEX (cilia-APEX2 and cilia-DAAO) expressing cells with desthio-biotin tyramide (DTBT) and D-Met for 30 min. For APEX2 proximity labeling cilia-iAPEX expressing wild-type and cilia-APEX2 expressing $Cep164^{-/-}$ cells (control) were pre-incubated with DTBT for 30 min, followed by 3 min $H_2O_2$. After cell lysis, labeled proteins were isolated by streptavidin chromatography and competitively eluted with biotin. Input, Unbound and Eluate fraction analysis by SDS-PAGE and western blotting. For mass spectrometric analysis, eluted proteins were digested in-solution using trypsin, peptides labeled with tandem mass tags (TMTs) and fractionated offline via reverse-phase chromatography. Quantitative proteomics was performed using LC-MS[3]: peptides were selected (MS[1]), fragmented for identification (MS[2]), and TMT reporter ions quantified (MS[3]). **b** Western blot analysis after proximity labeling from IMCD3 cells, as outlined in (**a**) (n = 2 independent experiments). IMCD3 $Ift88^{-/-}$ cell lysate served as antibody specificity and untreated control. SDS-PAGE and western blot analysis of Input and Eluate samples using indicated antibodies. Input 0.063%, Eluate 8.5%. **c** Volcano plot of statistical significance *versus* protein enrichment in cilia-APEX2 (left) and cilia-iAPEX (right) compared with control samples. Calculated *p* values (unpaired two-sided Student's *t* test) were plotted against TMT ratios for 5982 proteins. Gray and red circles indicate identified and known cilia proteins, respectively. Representative subunits of kinesin-2 (Kif3a), IFT-A (Ift122), IFT-B (Ift88) and the BBSome (Bbs4) are highlighted. Dotted lines indicate TMT ratios of $2^3$. **d** Selected clusters of two-way hierarchical cluster analysis of IMCD3 cilia-iAPEX proteome show known cilia proteins and highest scoring candidate cilia proteins. Legend shows relative protein abundances (in %). Full cluster analysis shown in Supplementary Fig. 4. **e** GO term enrichment analysis of protein clusters in (d) shows enrichment of ciliary categories. *p* values were calculated by one-sided Fisher's exact test. **f** Selected clusters with proteins identified after $H_2O_2$-mediated cilia-APEX2 proximity labeling. **g** GO term enrichment analysis of protein clusters in (**f**) identified enrichment of non-ciliary categories in $H_2O_2$ treated samples. *p* values were calculated by one-sided Fisher's exact test.

specific localization of both enzymes, exogenous $H_2O_2$-based APEX2 proximity labeling resulted in significant background biotinylation, which explains previous limitations in performing cilia proteomics with these cell lines (K. Hilgendorf and D. Mick, personal communication). In contrast, iAPEX labeling using D-Met led to specific biotinylation within primary cilia with reduced background in our fluorescence microscopy setup. SDS-PAGE and western blot analysis confirmed immense background biotinylation when using $H_2O_2$ (Fig. 7c, lanes 13-16), whereas D-Met-based labeling reduced the biotinylation to levels observed in the cilia-iAPEX IMCD3 cell line (Fig. 7c, lanes 9-12). While determining the primary cilia proteomes of both C2C12 and 3T3-L1 cell types brings additional challenges due to the relatively low ciliation rates[59–63] and the resulting low amounts of labeled cilia proteins, we are confident that our iAPEX workflow will finally enable mass spectrometry-based characterization of the cilia proteome during dynamic cilia processes, such as cell differentiation.

To further demonstrate the versatility of the iAPEX system, we expressed both transgenes in vivo in developing *Xenopus laevis* embryos by mRNA injections (Fig. 8a). During early embryonic development we found high enrichment of cilia-APEX2 in cilia of diverse tissues, such as primary cilia in the neural tube (Fig. 8b, c) and the multiciliated cells of the epidermis (Fig. 8d). To assess the functionality of the iAPEX enzymatic cascade in complex tissues, we investigated cilia protruding into the brain ventricular system at later embryonic stages and confirmed specific co-localization of both cilia-APEX2 and PKHD1$^{CTS}$-DAAO to cilia of multiciliated cells of the dorsal (Fig. 8a, e) and monociliated cells of the ventral hindbrain (Fig. 8a, f). Injection of the APEX substrate biotin tyramide into the ventricular system followed by $H_2O_2$ injection and incubation for 3 min led to robust biotinylation in cells expressing iAPEX enzymes (Fig. 8g). Importantly, incubation with D-norvaline (D-Nva), a non-proteogenic amino acid that does not affect neuronal function[64], was used to induce iAPEX labeling for 10 min, which led to specific biotinylation in multiciliated and monociliated cells (Fig. 8h, i), while no biotinylation was observed when biotin tyramide alone was injected and incubated for up to 30 min (Supplementary Fig. 7c–e). Moreover, cilia-iAPEX labeling was achieved in multiciliated (Supplementary Fig. 7f) and monociliated cells (Supplementary Fig. 7g) of dissected brains after fixation with paraformaldehyde, which underscores the versatility and suitability of the iAPEX system for future in situ, ex vivo and in vivo applications.

## Discussion

Proximity labeling technologies have emerged as powerful tools, especially for subcellular proteomics. The principle of an enzymatic activity that labels targets in its proximity brings three major advantages: 1) structures that can be marked by transgenes but cannot be purified by other means become accessible to proteomic investigation;

2) proximity labeling will capture low affinity and transient interactions[65]; 3) labeling with enzymatic activities allows for a high temporal resolution of proteomic analysis[2,22]. The latter is a particularly strong feature of ascorbate peroxidase (APEX), as its high enzymatic activity allows for sub-minute labeling times, which cannot be achieved by any other available methodology[66]. Yet, as the APEX activity requires hydrogen peroxide ($H_2O_2$), there had been limitations for a more general use, as the toxicity of externally added $H_2O_2$ limits its use in in vivo applications[67–69]. Moreover, the presence of endogenous peroxidases generates varying degrees of background signals that precludes specific labeling in many models -especially when studying structures of low abundance, such as the primary cilium (see Fig. 1a,b). The in situ APEX activation (iAPEX) enzymatic cascade solves this problem by locally producing $H_2O_2$ and thereby suppresses cell toxicity (see Fig. 2) and increases labeling specificity (see Fig. 3). While other proximity labeling methodologies remain powerful alternatives with individual strengths and weaknesses, the iAPEX technology still allows a very high temporal resolution of labeling, due to fast enzyme kinetics and the use of substrates that are bio-orthogonal (biotin tyramide) or very rare (D-amino acids) in most biological systems[70]. Importantly, the system is simple to use, as we can deliver both transgenes in one vector (see Figs. 1c, 7a), and the required substrates are commercially available at low costs.

Central to the iAPEX technology is the D-amino acid oxidase (DAAO) from *Rhodotorula gracilis*, which has been used in a wide variety of in vivo and ex vivo applications[35,64,71]. DAAO oxidizes a broad spectrum of D-amino acids in an FAD-dependent manner. Re-oxidation of FAD results in reduction of molecular oxygen ($O_2$) to $H_2O_2$ —the desired product— that is rapidly reduced by cellular peroxins, while the oxidized imino acids are metabolized to keto-acids and ammonium ($NH_3$)[29]. This molecular mechanism also points at potential limitations of the iAPEX system, which requires molecular oxygen and might cause metabolic imbalances. However, most tissues harbor much higher keto acid and $NH_3$ concentrations than DAAO-generated $H_2O_2$[64,72]. In the context of a cellular substructure, such as the primary cilium, we have noticed very low levels of $H_2O_2$ production (see Fig. 2) with minimal impact on biological processes in the final steps of sample preparation. Even though it is expected that the local concentrations of $H_2O_2$, keto acids and $NH_3$ produced by DAAO will be significant, we did not observe an obvious effect on intraflagellar transport in the timeframe of iAPEX labeling (see Supplementary Movies 1–3). While we cannot rule out local disruptions of other processes, we believe this will only have a minor impact on the technological application of iAPEX, as labelings are performed for relatively short amounts of time (5–30 min), right before fixation or lysis of the specimen. Importantly, protein oxidation did not affect protein identification or quantification by mass spectrometry (see Supplementary Fig. 4d).

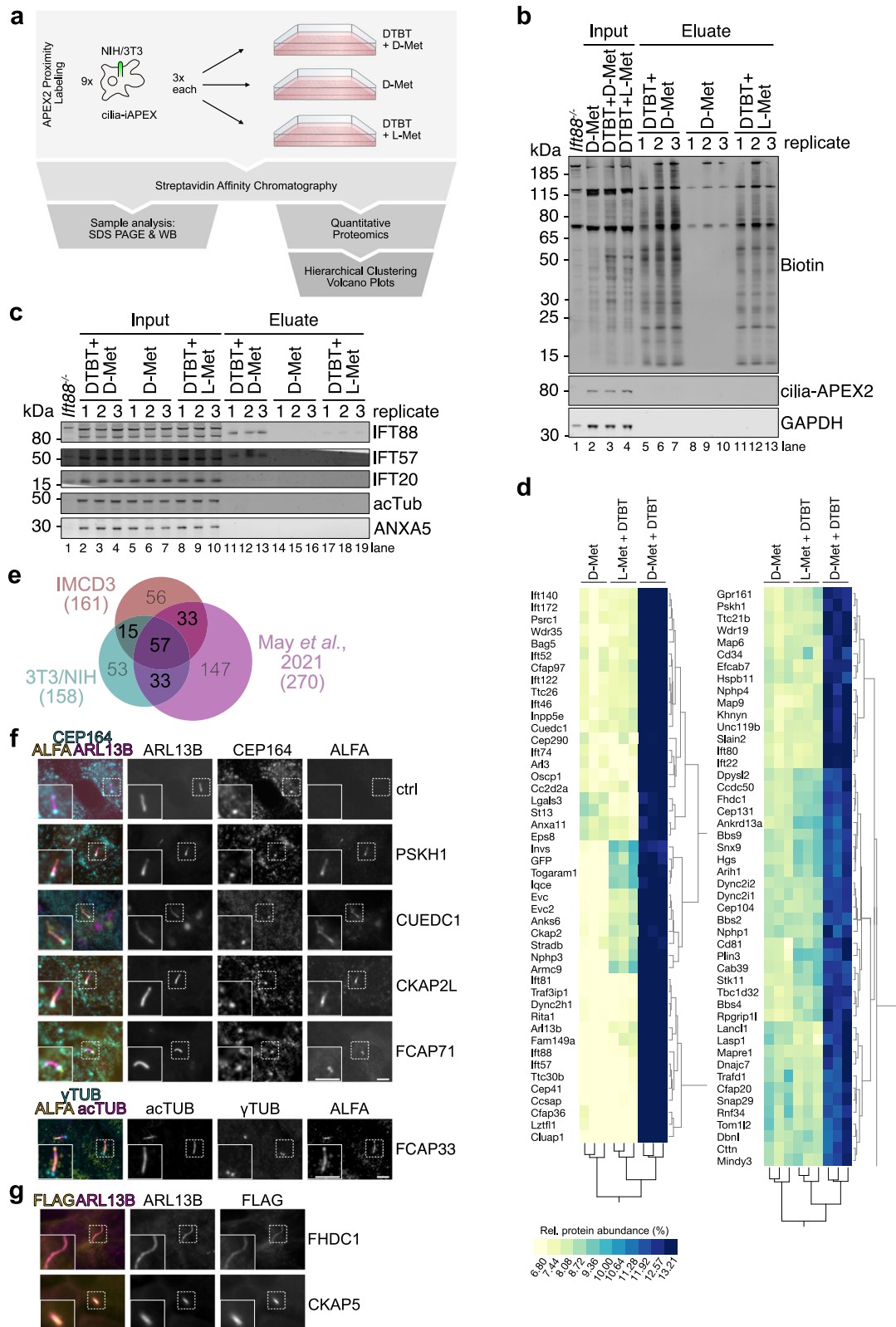

O$_2$ availability is expected to affect DAAO activity, particularly when investigating tissues with low oxygenation in in vivo applications. Optimized DAAO variants, such as mDAAO that has a -10fold higher affinity for O$_2$, may solve these issues but may also cause toxicity in tissues with high oxygen dependence[73,74]. Indeed, the more relevant limitation of the system we see is cellular availability of D-amino acids,

which have to traverse the plasma membrane via amino acid transporters and undergo cell type-specific metabolism[39,40,74–76]. While the D-amino acids can in principle cross both mitochondrial membranes (see Fig. 6b), this needs to be tested for other organelles. Cell type-specific differences can be expected, and D-amino acid concentrations and incubation times likely require optimization, as exemplified in our

**Fig. 5 | iAPEX allows specific quantitative primary cilia mapping in NIH/3T3 cells. a** Scheme of cilia-iAPEX proximity labeling workflow for primary cilia proteomics in NIH/3T3 cells. DAAO-dependent proximity labeling was performed by incubating cells with desthiobiotin tyramide (DTBT) and D-Met. As controls, cells were incubated with DTBT and L-Met, or D-Met only. Sample processing according to schematic in Fig. 4a. **b, c** After proximity labeling, cells were lysed, biotinylated proteins enriched by streptavidin chromatography, and Input and Eluate samples analyzed by SDS-PAGE and western blotting. **b** DTBT incubation causes background biotinylation ($n = 2$ independent experiments). Biotin was detected using fluorescently labeled streptavidin, cilia-APEX2 by GFP-specific antibodies. Input 0.063 %, Eluate 1.2 %. **c** Cilia proteins were specifically isolated after cilia-iAPEX labeling ($n = 2$). Indicated proteins were detected using specific antibodies. IMCD3 *Ift88*[-/-] cells served as an antibody control. Input 0.063 %, Eluate 8.8 %. **d** Two-way hierarchical cluster analysis of NIH/3T3 cell cilia-iAPEX proteome. Zoom on clusters highly enriched in cilia proteins. Full cluster analysis shown in

Supplementary Fig. 5. **e** cilia-iAPEX proteomes of IMCD3 and NIH/3T3 cells show distinct overlap with cell specific differences. Venn diagram depicting proteomic overlap of iAPEX-proximity labeled IMCD3 and NIH/3T3 cells against the cilia-APEX2 proteome[17]. **f, g** Validation of primary cilia localization of proteins previously not linked to cilia. **f** The representative immunofluorescence micrographs illustrate IMCD3 cells stably expressing both cilia-APEX2 (as localization control) and indicated primary cilia candidate proteins C-terminally fused to an ALFA-tag or ALFA-tag fusion alone (for FCAP33) ($n = 3$). Upon fixation, primary cilia and basal bodies were visualized using antibodies targeting ARL13B or acTub (cilium) and CEP164 or γTub (basal body) respectively, while the proteins of interest were stained with anti-ALFA antibodies. **g** Micrographs show NIH/3T3 cells transiently transfected with plasmids expressing CKAP5-GFP or FHDC1-FLAG ($n = 1$). Cells were fixed and primary cilia stained using indicated antibodies. FHDC1-FLAG was detected using anti-FLAG antibody, CKAP5-GFP by fluorescence. Scale bars = 5 μm in all panels.

study. Similar limitations in the uptake of substrates for proximity labeling, however, must be considered for all available technologies[2,77].

Our study demonstrates that due to varying levels of endogenous peroxidases, cell types will show different background profiles, which can have a detrimental impact for proteomics of small subcellular structures. Our data also show that even without $H_2O_2$ generation by DAAO, prolonged incubation with tyramides results in biotinylation, which can be attributed to varying levels of endogenous $H_2O_2$. Such background can be revealed by genetic controls, such as cell lines that express mislocalized enzymes (see Supplementary Fig. 1a) or lack the entire structure of interest –such as cilia-less *Cep164*[-/-] cells. However, generating genetic control cell lines is also flawed by potential genetic drift leading to proteomic alterations[78–80]. While isogenic control cell lines might still be the gold standard, we could show that a single iAPEX cell line can be employed to determine the cilia proteome of NIH/3T3 cells. Including information from samples with and without the $H_2O_2$ production by DAAO improved clustering resolution and circumvented the need for additional genetic controls –a major advantage for future in vivo applications.

In this proof-of-concept study we utilized the increase in sensitivity and specificity of the iAPEX technology to study the primary cilium proteome. A systematic comparison allowed us to identify several false-positive candidate cilia proteins from our own and other previous studies[13,14,17], such as proteins with known function in the endoplasmic reticulum or mitochondria. Despite a common realization that many proteins exist at multiple subcellular locations[81], it appeared unlikely that all previous hits fulfilled additional functions in primary cilia, and we now provide experimental evidence that they represented unspecific background in previous studies. In addition, we could confirm cilia localization of several candidate proteins with so-far unknown functions. PSKH1 is a putative serine protein kinase, mutations of which have recently been reported to cause hepatorenal ciliopathy in humans[82]. Interestingly, select ciliopathy mutations have been reported to render the enzyme catalytically inactive. Given our finding that the enzyme localizes to primary cilia, it will be interesting to identify the targets of PSKH1 in cilia and how it regulates cilia function. Similarly, biallelic null mutations in FCAP71/C19orf44 have recently been identified in patients with retinal degeneration[83]. Interestingly, with CKAP2L and CKAP5 we add two additional microtubule-associated proteins (MAPs) to the primary cilia proteome. After TOGARAM[84], CKAP5/XMAP215 is the second TOG-domain containing protein in cilia. CKAP5 was reported to polymerize microtubules and to bind to microtubule tips[85]. We identified TOGARAM and the non-TOG MAP CKAP2 in both IMCD3 and NIH/3T3 cilia. In contrast, CKAP5 appears to be NIH/3T3-specific, while CKAP2L appears IMCD3-specific. It will be exciting to investigate the potential function of these MAPs in primary cilia and how they might change the properties of cilia in the respective cell types.

By omitting $H_2O_2$ addition, iAPEX allows for much longer labeling times than with externally added $H_2O_2$, which not only reduces toxicity but also non-specific labeling effects at extended labeling times, thereby, leading to stronger biotinylation and increased sensitivity. This allowed us, despite the much smaller scale compared to previous studies[13,17], to identify candidate cilia proteins with unknown functions in cilia biology, not only in the well-studied IMCD3 model but also in previously inaccessible NIH/3T3 primary cilia, which had only been investigated using TurboID[18]. Here, we revealed an overlap of only about 45% between the cilia proteomes of these two cell lines. While the observed differences may support previous suggestions of cilia heterogeneity, they may also be caused by experimental variation in single mass spectrometry runs[86]. Nonetheless, our proof-of-concept study confirms that core cilia proteins, such as IFT, BBSome, kinesin and dynein subunits seem to be common to primary cilia of both cell types. We further identified many signaling components, such as kinases and putative transcription factors that differed between cell lines and could mediate unknown ciliary signaling pathways[7,87,88]. While differences in specialized cilia types, such as photoreceptor or olfactory sensory cilia have been well established[89,90], variations in the proteomes of primary cilia might explain specific clinical differences observed in ciliopathies[7,87,88]. With the iAPEX system available and the relative ease of use, we anticipate that many more cell type-specific cilia proteomes will become available soon to tackle this larger question in the cilia community.

The option to expand the iAPEX system to in vivo settings as explored here in *Xenopus* embryos will become a major advancement in detecting cilia proteome differences in specific cell types of an entire developing organism. The time-resolved nature of the iAPEX system will allow us to detect adaptations of cilia components in dynamically changing cells during embryonic development, with a focus on the orchestration of signaling pathways during tissue patterning and morphogenesis. We demonstrate here that this is possible even in organs that are relatively inaccessible but highly relevant for human disease, i.e., the developing neural tube and brain. Once substrate accessibility (tyramides, D-AAs and $O_2$) for the respective tissue has been optimized, cilia proteomics is mainly limited by the available material. In the case of *Xenopus* this will likely require the use of hundreds of embryos and the creation of transgenic lines expressing the iAPEX enzymes to study cilia proteome dynamics and its relevance for healthy development in a model organism in vivo. Similar strategies combined with tissue-specific expression of cilia-iAPEX will be powerful tools to investigate tissue- and cell type-specific defects in ciliopathy models.

At the same time, we also want to highlight the primary cilium as an interesting model to study enzyme reactions in a defined cellular microdomain. By using cilia targeting signals, we directed enzymes into the primary cilium and reconstituted a DAAO-APEX enzymatic

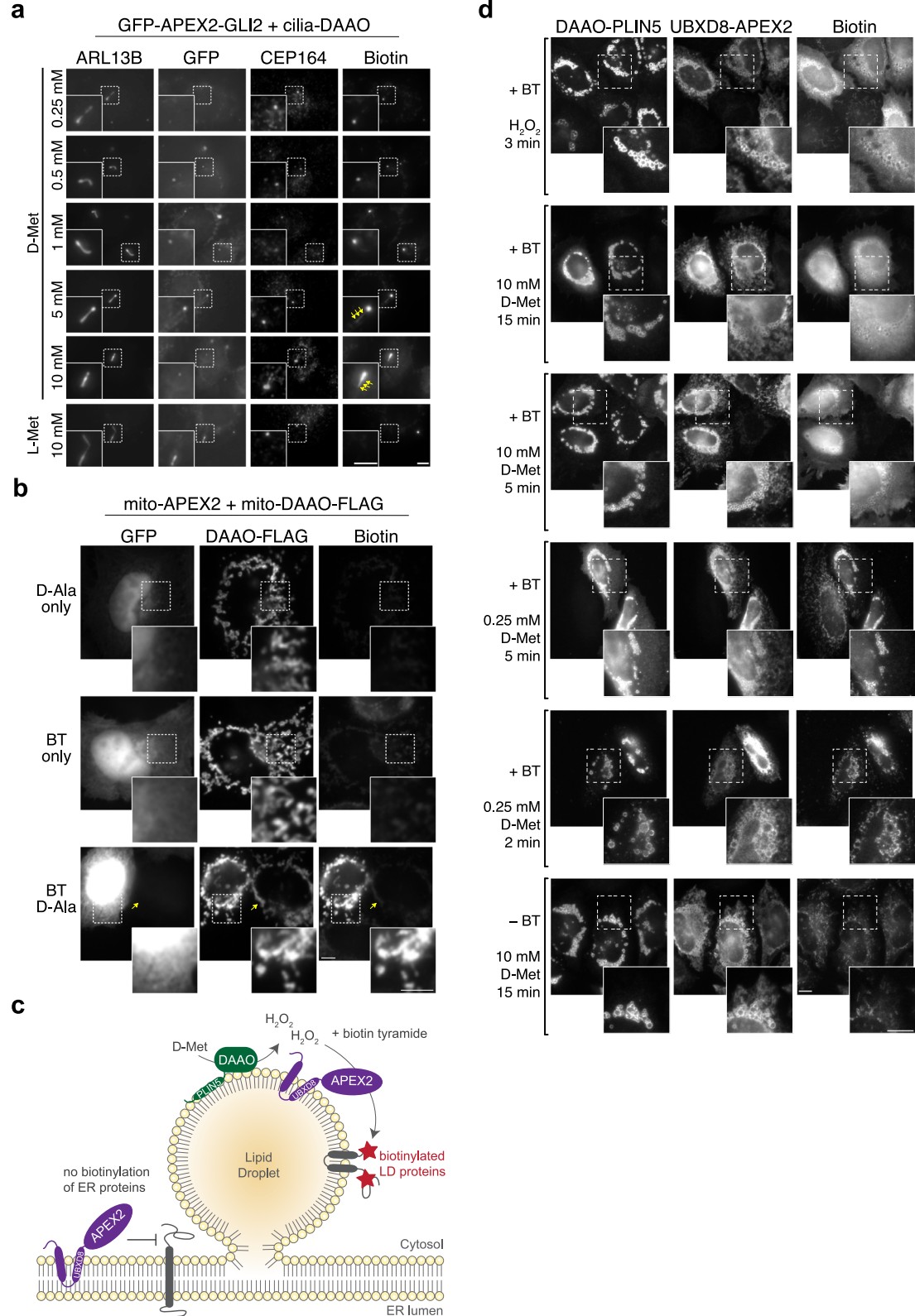

cascade, which can be studied in detail (substrate and product concentrations, reaction times and temperatures etc.) in a unique *in cellulo* environment. The primary cilium may therefore also be recognized as a "living test tube" with specific geometry and unique biophysical properties that may be a powerful model for future synthetic biology applications.

While we focused on the primary cilium as an example for a cellular microdomain that is difficult to purify, the iAPEX system can be applied to other cellular substructures[91,92], as exemplified by our lipid droplet (LD) and mitochondria applications (see Fig. 6). The iAPEX system requires two enzymatic activities to co-operate, which opens possibilities for proximity labeling of subpopulations of proteins or

**Fig. 6 | iAPEX enables locally restricted biotinylation of cilia tips, mitochondria and lipid droplets. a** Immunofluorescence micrographs of IMCD3 cells stably expressing GFP-APEX2-GLI2 and cilia-DAAO were immunostained after iAPEX labeling. 24 h post serum starvation, cells were labeled with 500 µM biotin tyramide (BT) and the indicated concentrations of D-/L-Met for 15 mins (n = 2 independent experiments). Cilia and ciliary base were visualized by anti-ARL13B and anti-CEP164 antibodies, respectively. Biotin was detected by fluorescently labeled streptavidin, GFP-APEX2-GLI2 by GFP fluorescence. Arrows indicate biotin signal along the ciliary shaft. Scale bars = 5 µm. **b** Immunofluorescence micrographs of iAPEX labeling in mitochondrial matrix of HeLa Kyoto cells transiently expressing COX4I1$^{1-26}$-APEX2 and soluble GFP from one plasmid (mito-APEX2) and SU9-DAAO-FLAG (mito-DAAO). 24 h after transfection, iAPEX labeling was performed by pre-incubating the cells with 500 µM BT for 30 min, followed by the addition of 10 mM D-Ala for 5 min (n = 1). mito-APEX2 transfected cells were visualized by GFP fluorescence (in cytoplasm and nucleus), and anti-FLAG antibodies stain mito-DAAO. Arrow highlights

cell transfected with mito-DAAO but without mito-APEX2, which shows no labeling. Scale bars = 5 µm. **c** Schematic representation of an iAPEX application to label the surface of lipid droplets (LDs). A DAAO-PLIN5 fusion on the surface of LDs converts D-amino acids into $H_2O_2$, which is used by LD-localized UBXD8-APEX2 to biotinylate LD proteins (red stars). UBXD8-APEX2 located in the endoplasmic reticulum (ER) cannot label nearby ER proteins. **d** Immunofluorescence micrographs of iAPEX labeling on LDs. HeLa Kyoto cells were transiently transfected with constructs expressing DAAO-ALFA-PLIN5 and S-UBXD8-APEX2. 24 h after transfection, LD formation was induced by adding 200 µM oleic acid / 0.2% BSA for 16 h. iAPEX labeling was performed by pre-incubating the cells with 500 µM BT for 30 min, followed by incubation with either $H_2O_2$ für 3 min or D-Met using the indicated concentrations and incubation times (n = 2). Cells were fixed and DAAO-ALFA-PLIN5 and S-UBXD8-APEX2 were visualized by anti-ALFA and anti-S-Tag antibodies, respectively. Scale bars = 10 µm.

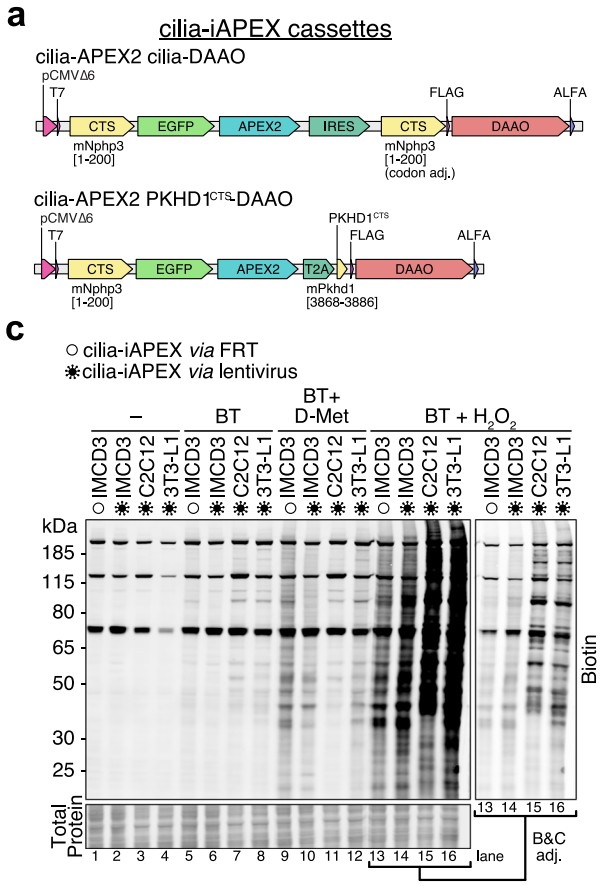

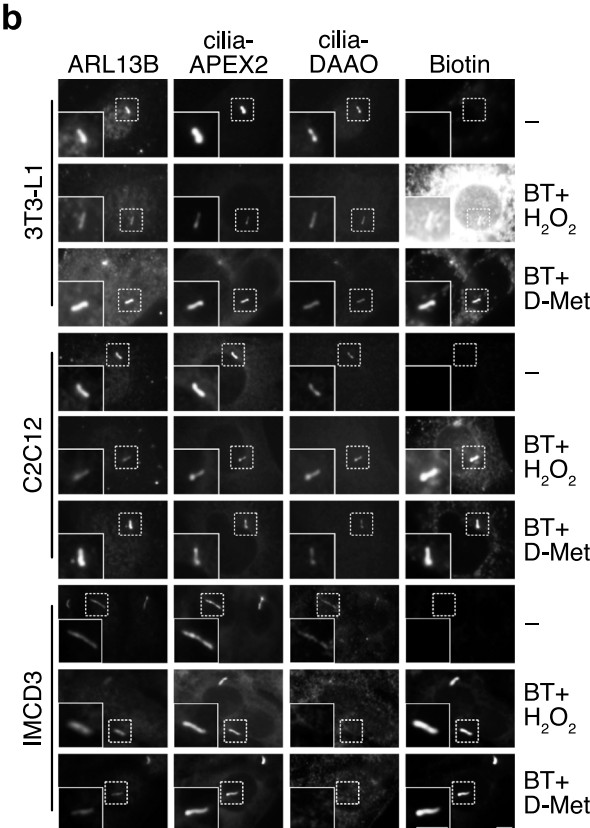

**Fig. 7 | Establishing iAPEX-based proximity labeling in previously inaccessible cell types. a** Polycistronic cassettes for cilia-iAPEX two-component expression. Transcription is controlled by low-expressing truncated CMV promoter ($P_{CMV\Delta6}$). Top, cilia-APEX2 and cilia-DAAO transgenes are separated by an internal ribosome entry site (IRES). Bottom, cilia-APEX2 and PKHD1$^{CTS}$-DAAO are separated by a T2A peptide. **b** 3T3-L1, C2C12 and IMCD3 cell lines have been infected with lentiviral vectors to express the cilia-iAPEX transgenes (depicted in (**a**)). Following proximity labeling with biotin tyramide (BT) and $H_2O_2$ or D-Met as indicated, cells were fixed and processed for immunofluorescence microscopy using antibodies specific to ARL13B to mark cilia, and ALFA tag to detect cilia-DAAO (n = 1). Biotinylation was

visualized using fluorescently labeled streptavidin and cilia-APEX2 by GFP fluorescence. Scale bars = 5 µm. **c** After incubation with indicated reagents for proximity labeling, IMCD3, C2C12, and 3T3-L1 cells expressing cilia-iAPEX after lentivirus infection (virus symbols) were lysed and analyzed by SDS-PAGE and western blotting (n = 1). cilia-iAPEX IMCD3 cells, in which the transgenes are expressed from the Flp-In locus served as control (empty circles). Biotin was visualized using fluorescently labeled streptavidin, equal protein loading (25 µg/lane) confirmed by total protein stain. Lanes 13-16 were brightness and contrast adjusted to visualize banding patterns. Where indicated BT and D-Met have been incubated for 30 min, $H_2O_2$ for 3 min.

structures that are defined by the co-localization of two markers. The use of different targeting signals for APEX and DAAO has the potential to increase spatial specificity to probe for subpopulations of organelles, proteins in complex with specific interaction partners, or organelle contact or biogenesis sites[91,92]. Similar co-localization-based experimental setups exist in the forms of split-APEX or split-TurboID,

where the enzymatic activities are reconstituted by complementation of two protein parts[93,94], and TransitID, which combines TurboID- with APEX2-based proximity labeling, followed by two-step purification schemes[95]. While the former requires two tagged proteins to directly interact in an orientation that allows APEX2 or TurboID reconstitution, which may induce non-physiological protein interactions, TransitID is

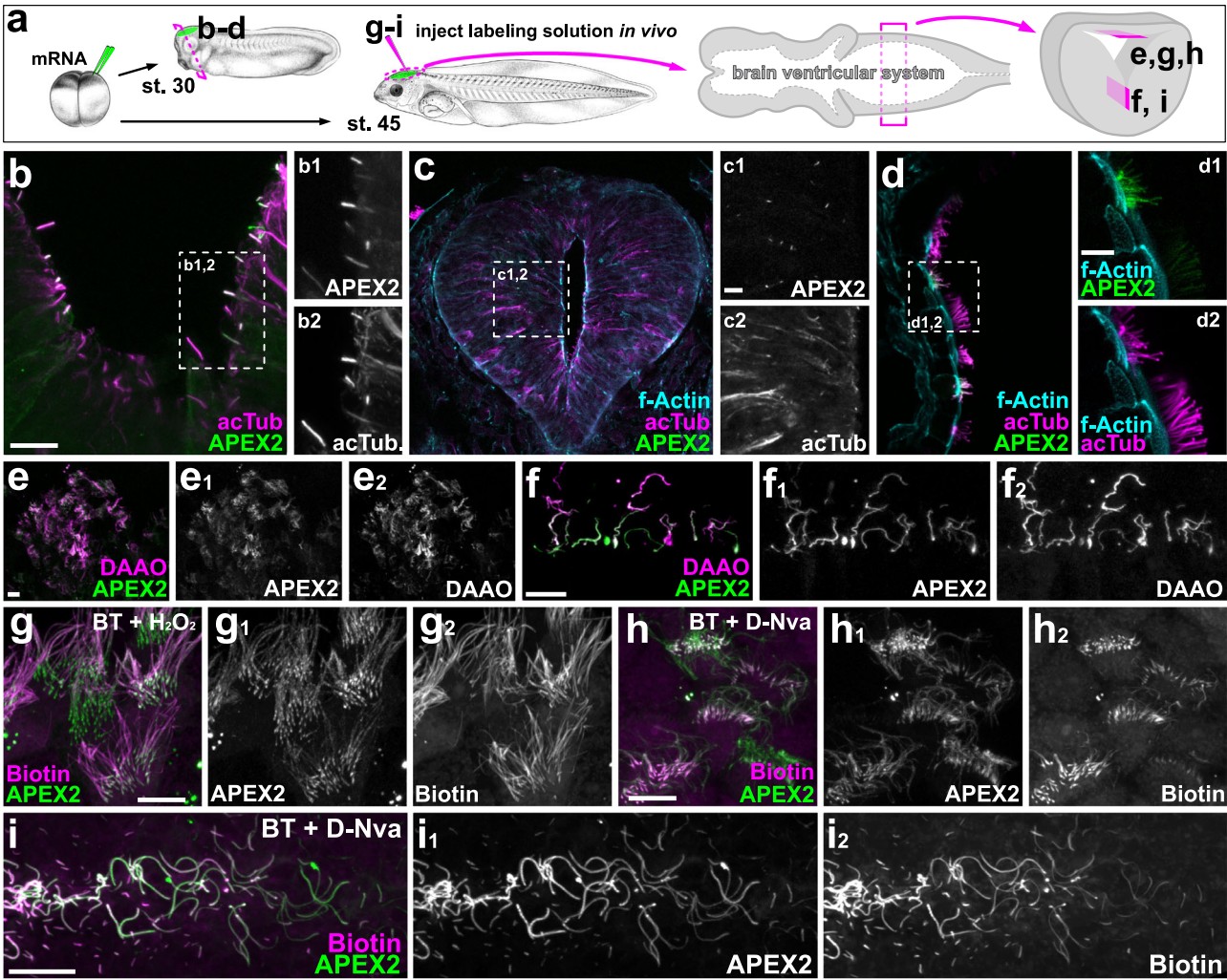

**Fig. 8 | cilia-iAPEX-based proximity labeling biotinylates primary and motile cilia in *Xenopus laevis*. a** mRNA transcribed from plasmids containing cilia-APEX2 or cilia-APEX2 and PKHD1^CTS-DAAO was injected into one or two dorsal animal blastomeres of four-to-eight-cell *Xenopus laevis* embryos to target constructs to the central nervous system. Embryos were reared to tailbud / tadpole stages (st. 30 / 45). For in vivo proximity biotinylation, labeling solutions were injected into the brain ventricular system at st. 45. Hemisections through the brain area (st. 30) or brain preparations (st. 45) were immunostained to visualize enzyme expression and biotinylation. **b–d** APEX2, expressed from cilia-APEX2 (**b**) or cilia-APEX2 PKHD1^CTS-DAAO (**c, d**) constructs, (detected by an anti-GFP antibody, shown in green) localizes to acetylated α-tubulin (acTub)-positive primary cilia (shown in magenta) of the ventral (**b**) and lateral neural tube (**c**) and to motile cilia of multiciliated epidermal cells (**d**). Representative micrographs of two independent experiments ($n = 2$) and six embryos ($n = 6$) are shown. **e, f** Co-localization of cilia-APEX2 (green) and cilia-DAAO (detected by anti-ALFA-tag antibody, magenta) expressed from cilia-APEX2 PKHD1^CTS-DAAO constructs in cilia of dorsal multiciliated (**e**) and ventral monociliated cells (**f**) in the tadpole hindbrain. $n = 4$, $n = 8$. **g–i** In vivo proximity labeling in st. 45 brains mediated by cilia-iAPEX enzymes, fluorescently labeled streptavidin detects biotin (in magenta) in dorsal multiciliated (**g, h**) and ventral monociliated (**i**) cells. Biotin tyramide (BT) and $H_2O_2$ injected sequentially (**g**), or BT and D-norvaline (D-Nva) injected together (**h, i**) into the hindbrain ventricle. Biotinylation was stopped using fixative containing sodium azide after three (**g**) or ten (**h, i**) minutes. $n = 1$, $n = 2$ for (**g**), $n = 2$, $n = 6$ for (**h, i**). Scale bars: 10 μm, *Xenopus* illustrations © Natalya Zahn (2022)[108].

limited by the relatively slow enzymatic kinetics of TurboID labeling. Moreover, both systems require exogenous addition of $H_2O_2$. Therefore, despite clear advantages, split-APEX and TransitID may not be suitable for biological systems with higher levels of endogenous peroxidases that increase background labeling (see Fig. 1a), or where biotin scavenging or $H_2O_2$ toxicity should be avoided[96,97]. Since both technologies rely on APEX2 activities, they could be combined with in situ generation of $H_2O_2$ by DAAO or similar enzymes. It remains to be seen whether these methods will be combined by many non-specialists, as this would further increase the complexity of experimental setups. In this regard, we believe that the iAPEX methodology provides a powerful improvement of an existing technology that is simple to implement and greatly increases sensitivity and specificity for subcellular proteomics applications.

## Methods

### Ethical statement

We confirm that our research complies with all relevant ethical regulations. All animals were treated according to the German regulations and laws for care and handling of research animals and experimental manipulations were approved by the Regional Government Stuttgart, Germany (RPS35-9185-81/0471 and RPS35-9185-99/426).

### Cell culture and cell line generation

Wild-type cells and established cell lines, including 3T3-L1 pre-adipocytes, C2C12 myoblasts, HEK293T cells, HeLa Kyoto cells, and NIH/3T3 fibroblasts, were cultured in DMEM (Fisher Scientific, Cat. No. 11594486). IMCD3 cells were grown in DMEM/F12 (Fisher Scientific, Cat. No. 11594426). All media were supplemented with 7.5% FBS (Fisher

Scientific, Cat. No. 11573397), except for 10% FBS (PAN Biotech) in HeLa cell medium. All cells were propagated at 37 °C, 5% $CO_2$. To induce ciliation, cells were serum-deprived in 0.2% FBS containing media for 24 h. Lipid droplet (LD) formation was induced by 200 μM oleic acid in complex with 0.2% BSA in growth medium for 16 h. IMCD3 cell lines stably expressing cilia-APEX2 and control-APEX2 have been previously described[17]. IMCD3 cell lines stably expressing cilia-iAPEX, cyto-iAPEX, and cilia-DAAO-APEX2, as well as NIH/3T3 cell line stably expressing cilia-iAPEX, were generated using the Flp-In system as previously described[98]. A Flp-In-compatible vector (pEF5B-FRT-cilia-APEX2/cilia-DAAO) with back-to-back CMVΔ6 and EF1α-TATA-box-mutant promoters was generated to enable simultaneous expression of genetically targeted APEX2 and DAAO transgenes. For cloning, DAAO was amplified from pAAV-SypHer2-DAAO-NES (gift from L. Prates Roma). Forward transfections were performed using jetPRIME transfection reagent (VWR, Cat. No. 101000046) according to manufacturer's guidelines. Cloning cylinders (Sigma-Aldrich, Cat. No. CLS31668) were employed to obtain cell clones.

For transient transfection of NIH/3T3, $5 \times 10^4$ cells were grown on round 12-mm #1.5 coverslips (Fisher Scientific, 11846933) in a 24-well plate. 500 ng of plasmid DNA was transfected into cells using jetPRIME transfection reagent (VWR, Cat. No. 101000046) according to manufacturer's guidelines. 4 hours post-transfection medium was changed to 0.2% FBS containing media for 24 h to induce ciliation. For transient transfection of Hela Kyto cells, $2 \times 10^4$ or $5 \times 10^4$ cells were grown on round 12-mm #1.5 coverslip in a 12- or 24-well plate, respectively. 500 ng of plasmid DNA was incubated with 2 μg of polyethylenimine (PEI) in $ddH_2O$ containing 150 mM NaCl for 20 mins. The mixture was added onto the cells in a dropwise manner. 24 hours post-transfection, LD formation was induced for 16 h.

For 3T3-L1, C2C12 and RPE-1 cells, the multicistronic lentiviral vector pLVX-cilia-APEX2-IRES-cilia-DAAO was designed, which contains a CMVΔ6 promoter and an internal ribosomal entry site (IRES) to enable co-expression of the transgenes cilia-APEX2 and cilia-DAAO. The multicistronic plasmid was synthesized by BioCat. Lentivirus was produced by transfecting HEK293T cells with second-generation lentiviral vectors (psPAX2, pMD2g-VSV-G, and pLVX-cilia-APEX2-IRES-cilia-DAAO in a 1:1:2 ratio). After 24 h, medium was replaced. 24 h later lentivirus containing supernatant was collected, filtered through 0.45 μg PES filter, and supplemented with 4 μg/ml polybrene for infection of IMCD3 and 3T3-L1 and 10 μg/ml for C2C12 cells. Infected cells were first propagated, then sorted by fluorescence-activated cell sorting (FACS) based on GFP expression. All cell lines were verified by immunofluorescence (IF) microscopy and western blotting (WB) using protein tag specific antibodies.

## APEX2 proximity labeling

For conventional APEX-based proximity labeling, cells were incubated with 0.5 mM APEX substrate, biotin tyramide or desthiobiotin tyramide (BT, Iris Biotech, Cat. No. LS-3500; or DTBT, Iris Biotech, LS-1660; respectively), for 30 min at 37 °C before addition of $H_2O_2$ (Sigma Aldrich, Cat. No. H1009) to a final concentration of 1 mM and incubation at RT for 3 min. For DAAO-dependent ("iAPEX") labeling, APEX substrate was added to the cells together with 10 mM D-amino acid at 37 °C for 30 min, unless noted otherwise. Unlabeled samples were kept untreated. After substrate incubation, the medium was aspirated, and cells washed three times with quenching buffer (1× PBS containing 10 mM sodium ascorbate, 10 mM sodium azide, and 5 mM Trolox). Cells grown on glass coverslips for IF microscopy were fixed immediately. Samples intended for proteomic and WB analyses were prepared by lysing and scraping the cells off the growth surface in ice-cold lysis buffer (0.5% [vol/vol] Triton X-100, 0.1% [wt/vol] SDS, 10% [wt/vol] glycerol, 300 mM NaCl, 100 mM Tris/HCl, pH 7.5, and protease inhibitors) containing 10 mM sodium ascorbate, 10 mM sodium azide, and 5 mM Trolox. The collected lysate was briefly vortexed, incubated on

ice for 15 min, and cleared by centrifugation (20.000 × $g$ for 30 min at 4 °C).

## $O_2$ consumption analysis

$1.5 \times 10^4$ IMCD3 cells were seeded into a 96-well XF cell culture microplate (Agilent, Cat. No. 103794-100) in 80 μl of growth medium and left to settle at RT for 1 h. Cells were grown overnight at 37 °C in a 5% $CO_2$ incubator, followed by serum starvation for 24 h. The sensor cartridge (Agilent, 103793-100) was prepared following the manufacturer's instructions.

On the day of the assay, Seahorse XF Assay Medium (pH 7.4, Agilent, Cat. No. 103575-100) was prepared by adding Seahorse XF glucose (f.c. 17.5 mM, Agilent, Cat. No. 103577-100), pyruvate (f.c. 1 mM, Agilent, Cat. No. 103578-100) and L-glutamine (f.c. 2 mM, Agilent, Cat No. 103579-100). The cells were washed twice with 100 μl of prewarmed XF Assay Medium. Finally, 180 μl of XF Assay Medium was added to each well. The plate was incubated at 37 °C in a non-$CO_2$ incubator for 60 minutes before starting the assay. Oligomycin (15 μM stock, f.c. 1.5 μM, Sigma-Aldrich, Cat. No. 4876), D- or L-amino acids (100 mM stock, f.c. 10 mM), and Hoechst (100 μM stock, f.c. 10 μM) were loaded into individual injection ports of the sensor cartridge. If a chemical was to be omitted during injection, medium was added to the designated port instead. The microplate and sensor cartridge were loaded into the Agilent Seahorse Analyzer. The experimental protocol included an initial calibration and equilibration step, and measurement cycles consisting of 3 min mixing, 15 s wait period and 3 min of measurement. $O_2$ consumption rates (OCR) were measured for 20 cycles. Oligomycin and amino acids were injected after 4 and 8 cycles, respectively. After 20 cycles Hoechst staining was performed for 3 min.

Wells with initial OCR values (Y1 rate) > 20 pmol/min, initial $O_2$ levels (Y1 level) ≈ 100 mmHg reducing to ≤ 20 mmHg after oligomycin, and initial pH near 7.4 were deemed acceptable, while those failing to meet these criteria were flagged and excluded during data analysis using the software's plate map modification tool. Post-assay, the XF cell culture microplate was transferred to the BioTek Cytation system to allow normalization of each well to the respective cell number.

## Streptavidin affinity chromatography

Desthiobiotin tyramide labeled lysates were prepared as input for chromatography by adjusting them to equal concentrations and volumes. Fractions of input samples were taken as SDS-PAGE and WB controls. Streptavidin Sepharose High Performance Medium (Cytiva, 17-5113-01) was washed, equilibrated and then incubated with lysates at RT under rotation for 1 h. Unbound material was collected from settled beads and kept for WB analysis. Loaded beads were washed extensively with lysis buffer and spun dry before elution. Competitive elution buffer (100 mM Tris/HCl pH 7.5, 5 mM biotin) was added to the beads and incubated for 30 min shaking in a thermomixer at 950 rpm at RT. Elution was repeated and the eluates were combined. Eluates were subjected to centrifugal filter unit (Amicon Ultra, 0.5 ml 30 K, Sigma-Aldrich, Cat. No. UFC5030) to be concentrated before mass spectrometric sample preparation. 10% of each eluate was kept for WB analysis.

## Mass spectrometry

For mass spectrometry, $12 \times 10^6$ cells were seeded per 500 $cm^2$ plate, grown for three days and starved for 24 h before APEX labeling and streptavidin chromatography. Mass spectrometry was performed at the EMBL Proteomic Core Facility in Heidelberg, Germany. For the mass spectrometric analysis, Amicon-concentrated eluates were subjected to an in-solution tryptic digest, following a modified version of the Single-Pot Solid-Phase-enhanced Sample Preparation (SP3) technology[99,100]. 20 μl of a slurry of hydrophilic and hydrophobic Sera-Mag Beads (Thermo Scientific, #4515-2105-050250, 6515-2105-050250) were mixed, washed with water and were then reconstituted in 100 μl

water. 5 µl of the prepared bead slurry were added to 50 µl of the eluate following the addition of 55 µl of acetonitrile. All further steps were prepared using the King Fisher Apex System (Thermo Scientific). After binding to beads, beads were washed three times with 100 µl of 80% ethanol before they were transferred to 100 µl of digestion buffer (50 mM HEPES/NaOH pH 8.4 supplemented with 5 mM TCEP, 20 mM chloroacetamide (Sigma-Aldrich, #C0267), and 0.25 µg trypsin (Promega, #V5111). Samples were digested over night at 37 °C, beads were removed, and the remaining peptides were dried down and subsequently reconstituted in 10 µl of water. 80 µg of TMT10plex (Thermo Scientific, #90111)[101] label reagent dissolved in 4 µl of acetonitrile were added and the mixture was incubated for 1 h at room temperature. Excess TMT reagent was quenched by the addition of 4 µl of an aqueous solution of 5% hydroxylamine (Sigma, 438227). Mixed peptides were subjected to a reverse phase clean-up step (OASIS HLB 96-well µElution Plate, Waters #186001828BA). Peptides were subjected to an offline fractionation under high pH conditions[100]. The resulting 12 fractions were analyzed by multistage mass spectrometry (MS$^3$) on a Orbitrap Fusion Lumos system (Thermo Scientific).

To this end, peptides were separated using an Ultimate 3000 nano RSLC system (Dionex) equipped with a trapping cartridge (Precolumn C18 PepMap100, 5 mm, 300 µm i.d., 5 µm, 100 Å) and an analytical column (Acclaim PepMap 100. 75 × 50 cm C18, 3 mm, 100 Å) connected to a nanospray-Flex ion source. The peptides were loaded onto the trap column at 30 µl per min using solvent A (0.1% formic acid) and eluted using a gradient from 2 to 80% Solvent B (0.1% formic acid in acetonitrile) over 2 h at 0.3 µl per min (all solvents were of LC-MS grade). The Orbitrap Fusion Lumos was operated in positive ion mode with a spray voltage of 2.5 kV and capillary temperature of 275 °C. Full scan MS spectra with a mass range of 375–1.500 m/z were acquired in profile mode using a resolution of 120.000 (maximum fill time of 50 ms; AGC Target was set to Standard) and a RF lens setting of 30%. Ions were selected in the quadrupole applying an isolation window of 1.5 m/z. Fragmentation was triggered by HCD using a collision energy of 36%. Ions were analyzed in the ion trap (maximum fill time of 50 ms; AGC target was set to Standard). The top 5 precursors were selected by synchronous precursor selection (SPS) between 400 to 2.000 m/z with a precursor ion exclusion width of −18 and +5 m/z. Their fragmentation was triggered by HCD using a collision energy of 70%. Ions were analyzed in the Orbitrap using a resolution of 50.000 (maximum injection time was set to 105 ms and the AGC Target was set to Custom and 200%).

Acquired data were analyzed using FragPipe[102] and a UniProt *Mus musculus* FASTA database (UP000000589, ID10090 with 21.968 entries, date: 27.10.2022, downloaded: January 11th 2023) including common contaminants. The following modifications were considered: Carbamidomethyl (C, fixed), TMT10plex (K, fixed), Acetyl (N-term, variable), Oxidation (M, variable) and TMT10plex (N-term, variable). The mass error tolerance for full scan MS spectra was set to 10 ppm and for MS/MS spectra to 0.02 Da. A maximum of 2 missed cleavages were allowed. A minimum of 2 unique peptides with a peptide length of at least seven amino acids and a false discovery rate below 0.01 were required on the peptide and protein level[103].

## MS data analysis

Obtained TMT reporter intensities in one experiment were preprocessed using Perseus (version 2.0.11.0). Raw TMT signal intensities were log$_2$ transformed, filtered for valid values in at least two out of three replicates in each group and missing values were replaced from normal distribution (width: 0.3; standard deviation down shift: 1.8). The filtered and imputed data was then Z-score normalized, where the mean of each column (reporter intensities for all proteins) was subtracted from each value and the result divided by the standard deviation of the column. Hierarchical cluster analyses were performed on the preprocessed and normalized data according to Ward's

minimum variance method using two-way unstandardized clustering in JMP software (Statistical Analysis System; v17.2.0). Candidates with $p$ values > 0.05 were not displayed in clusters.

Gene Ontology enrichment analysis was performed using the web-based tool EnrichR (https://maayanlab.cloud/Enrichr/). Protein lists to be investigated (such as proteins in subclusters) were compared to all proteins identified in the respective mass spectrometry experiment as background list[104]. Scatter plots displaying Pearson correlation coefficients were generated using Perseus.

## Expansion microscopy

Ultrastructure expansion microscopy (U-ExM) was performed following a modified protocol by Gambarotto et al.[105] to achieve high-resolution imaging of physically magnified cellular structures. $5 × 10^4$ cells were seeded per well on round 12-mm #1.5 coverslips (Fisher Scientific, 11846933) in 24-well plates. APEX proximity labeling was performed as described. After quenching, 300 µl cross-linking solution (1.4% formaldehyde (Sigma-Aldrich, F8775) / 2% acrylamide (Sigma-Aldrich, A4058) in 1× PBS) was added to each well, and cells were incubated for 5 h at 37 °C. For gelation, 90 µl monomer solution mixed with 5 µl 10% APS (Thermo Fisher, 17874) and 5 µl 10% TEMED (Thermo Fisher, 17919) was drop-wise applied on parafilm placed in an ice-cold humid chamber. For 1 ml of monomer solution 500 µl of sodium acrylate (Sigma-Aldrich, 408220, 38% stock in nuclease-free water, f.c. 19%), 250 µl acrylamide (Sigma-Aldrich, A4058, 40% stock, f.c. 10%), 50 µl N,N′-methylenbisacrylamide (Sigma-Aldrich, M1533, 2% stock, f.c. 0.1%) and 100 µl 10× PBS were mixed. Coverslips were positioned cell-side down on the gelation solution, incubated on ice for 5 min, then at 37 °C for 1 h. Gels were detached by transferring coverslips to 6-well plates with 1 ml denaturation buffer (200 mM SDS, 200 mM NaCl, 50 mM Tris/HCl in water, pH 9) for 15 min at RT, then incubated in 1.5 ml reaction tubes with fresh denaturation buffer at 95 °C for 1.5 h. For expansion, gels were transferred to individual 250 ml beakers with dH₂O and agitated at RT. Every 10 min water was replaced for three times. After measuring expansion, gels were shrunk in 1× PBS for 30 min and cut into quarters. For antibody staining, gel pieces were incubated in 200 µl of primary antibody diluted in 2% BSA (Carl Roth, 3737.3) in 1× PBS at 4 °C overnight. The next day, gels were washed with PBS + 0.1% Tween-20 (Carl Roth, 9127.2), then incubated in 200 µl of secondary antibody in 2% BSA in 1× PBS for 2.5 h at 37 °C with agitation, protected from light. After three washes gels were re-expanded in dH₂O with 0.02% sodium azide and incubated at 4 °C for three days.

25 mm coverslips were coated with poly-D-lysine (100 µg/ml (Gibco, A3890401) overnight at 4 °C, rinsed, dried, and stored. Imaging was conducted on a Zeiss LSM900 confocal microscope with Airyscan using a 63× oil objective (Plan-Apochromat 63x/1.40 Oil DIC M27), with the convex gel side placed against the coated coverslip surface in a round imaging chamber.

## Immunofluorescence microscopy

For immunofluorescence microscopy, all steps were performed at RT. $5 × 10^4$ cells were grown on round 12-mm #1.5 coverslips (Fisher Scientific, 11846933) in a 24-well plate and fixed in 4% PFA for 15-20 min and permeabilized with 0.1% Triton X-100 for 10 min. Cells were rinsed three times with 1× PBS between subsequent steps. Cells were incubated in blocking buffer (3% BSA, 5% serum, in 1× PBS) for 30 min. Primary and secondary antibody dilutions were prepared in blocking buffer. Primary antibody mixes were incubated on cells for 1 h, secondary antibody mixes for 30 min. The latter contained fluorescent secondary antibodies or streptavidin. DNA was stained with Hoechst 33258 (1:1000 in 1× PBS, Thermo Scientific, H3570). Lastly, coverslips were mounted on glass slides using Mowiol or Fluoromount G mounting medium (Fisher Scientific, 15586276). APEX2 enzyme was detected by GFP fluorescence or S-Tag antibodies, DAAO and fusion enzymes were detected by FLAG or ALFA staining.

Prepared specimens were imaged on Leica DMi8 microscope (LAS X software, version 3.7.0.20979) with PlanApochromat oil objectives (63×/1.4 NA) using appropriate filters. Images were captured using a Leica DFC3000 G camera system. For lipid droplet experiments, cells were imaged using a Zeiss Axio Observer Z1 inverted microscope equipped with a Colibri 7 LED light source, a PlanApochromat oil objective (63×/1.4 NA) and appropriate filter sets. 15 z-sections with 240 nm intervals were collected using a Rolera EM-C2 camera (QImaging). Images were processed using ImageJ2 (v2.14.0/1.54 f) or Fiji (v2.1.0/1.53c).

## Primary cilia intensity quantification

Fluorescence intensities were quantified by ImageJ plugin CiliaQ (v0.1.4)[106]. A cilia mask was obtained by applying the RenyiEntropy threshold algorithm to either the ARL13B or GFP channel of each image.

## Live-cell imaging of intraflagellar transport

$2.5 \times 10^5$ IMCD3 Flp-In cells were seeded onto round 25 mm #1.5 coverslips (Fisher Scientific, 10593054) in a 6-well plate. The following day, a plasmid for IFT38-GFP-APEX2 and PKHD1$^{CTS}$-FLAG-DAAO-ALFA expression was transfected into the cells using jetPRIME transfection reagent (VWR, Cat. No. 101000046) according to manufacturer's guidelines. 4 hours post-transfection medium was changed to 0.2% FBS containing media for 24 h to induce ciliation. Cells were washed twice with 1× PBS and coverslips transferred to a round imaging chamber containing 1× PBS. Imaging was performed immediately using Leica DMi8 microscope (LAS X software, version 3.7.0.20979) with PlanApochromat oil objectives (63×, 1.4 NA) using the filter for 488 nm. Time-lapse videos were further processed in ImageJ2 (v2.14.0/1.54 f).

## Live-cell imaging of peroxidase-catalyzed Amplex™ UltraRed oxidation

$2.5 \times 10^5$ cells were seeded onto round 25 mm #1.5 coverslips (Fisher Scientific, 10593054) in a 6-well plate and serum-starved one day before imaging. Cells were imaged in live using either a Zeiss LSM800 with a 40x objective (Plan-Apochromat 40×/1.3 Oil DIC UV-IR M27) or a Zeiss LSM900 confocal microscope with a 63× oil objective (Plan-Apochromat 63×/1.40 Oil DIC M27), diode lasers 488 and 561 and appropriate filters. For imaging, the coverslip was placed in a round chamber and covered with cold (4 °C) HEPES-buffered DMEM/F12 medium without phenol red.

For live-cell time series of resorufin channel only, multi-channel images were taken before and after the Amplex UltraRed (AmUR) oxidation experiment. Resorufin channel recording was started before substrate addition to the cells and set to record as fast as possible. Cells were covered with 800 μl imaging buffer. AmUR (Fisher Scientific, 10737474) was pre-mixed with either D-Met or $H_2O_2$ to be added to the solution dome to a final concentration of 50 μM AmUR and 10 mM D-Met or 10 mM $H_2O_2$ respectively. Substrates and mixes were kept on ice.

For time-lapse imaging, 400 μl imaging medium was supplemented with 100 μM AmUR and 20 mM D-Met or 2 mM $H_2O_2$, then directly and carefully pipetted into the center of the 400 μl solution dome to yield halved final concentrations. Both GFP and resorufin channels were monitored at 4.8 s intervals over a total duration of 264 s.

## SDS-PAGE and western blotting

For SDS-PAGE and western blotting, standard techniques were applied. Cell lysates were generated as described before. 20 μg protein was separated on 4–12% Bis-Tris polyacrylamide gels (Invitrogen NuPAGE, WG1403BX10) and transferred onto nitrocellulose membrane (Fisher Scientific, 15269794) for fluorescence detection. Before blocking, membranes were dried and total protein stain (LI-COR, 926-11011) was used according to manufacturer's guidelines to verify equal gel loading and transfer. Destained or untreated membranes were blocked in Intercept (TBS) Protein-free blocking buffer (LI-COR, 927-80001) at RT for 30 min. Specific primary antibody mixes were prepared in 5% milk in 1× TBS and membranes were incubated at 4 °C overnight. Fluorescently coupled secondary antibodies were used to visualize stained proteins and were imaged on a LI-COR Odyssey CLx laser scanner.

## Xenopus methods

For expression of iAPEX enzymes, capped mRNA was synthesized from linearized plasmids using mMESSAGE mMACHINE kit (Invitrogen AM1344). Injection drop size was calibrated to 4 nl to deliver 300 pg of mRNA per single injection. Embryos were grown to desired stages and either fixed in 4% PFA in 1× PBS⁻ overnight at 4 °C for analysis of construct localization or treated for iAPEX-mediated biotinylation. For in vivo proximity labeling of ependymal cilia at st. 45, either a) biotin tyramide (0.5 mM) and $H_2O_2$ (1 mM) were sequentially injected into the ventricular system with an offset of 10 min, $H_2O_2$ was incubated for 3 min; or b) a labeling solution containing biotin tyramide (2 mM) and D-norvaline (40 mM) was injected into the ventricular system and incubated as indicated (up to 30 min). All brain injections were done with a drop size calibrated to 33.5 nl. For post-fixation labeling, embryos were PFA-fixed for 20 min at room temperature, washed in PBS, and dissected brains were incubated in a labeling solution containing biotin tyramide (500 μM) and D-norvaline (10 mM) for up to 10 min. All biotinylation reactions were terminated by fixation in 4% PFA in 1× PBS⁻ containing 10 mM sodium azide overnight at 4 °C. (Immuno-)fluorescent staining to detect iAPEX enzymes, biotinylation and cell markers was carried out on PFA-fixed embryos following dissection. In short, tissues were washed in PBS⁻, permeabilized in PBS⁻ containing 0.1 % Triton X-100 and pre-blocked in CAS blocking buffer (Thermo Fisher 008120). Primary and secondary antibodies, streptavidin and phalloidin were added in blocking buffer and incubated overnight at 4 °C. Embryos and brain tissues were washed in PBS⁻ and mounted in 0.5% low melting agarose or Mowiol, respectively, for imaging on a Zeiss LSM 700.

## Statistics & reproducibility

Statistical analyses were conducted using GraphPad Prism (10.3.1(464)) or JMP software (Statistical Analysis System; v17.2.0). No statistical method was used to predetermine sample size. No data were excluded from the analyses. The experiments were not randomized. The investigators were not blinded to allocation during experiments and outcome assessment.

## Miscellaneous

All graphs were prepared with GraphPad Prism (10.3.1(464)). Figures and schematics were prepared using Affinity Designer (1.10.8). Xenbase was the source of *Xenopus* illustrations (www.xenbase.org RRID:SCR_003280).

## Antibodies, plasmids, oligonucleotides and cell lines

All information can be found in Supplementary Data 4. Plasmids and full sequences are available upon request.

## Reporting summary

Further information on research design is available in the Nature Portfolio Reporting Summary linked to this article.

# Data availability

Data of this study are available in the Figures or the Supplementary Information. All proteomics data generated in this study have been deposited to the ProteomeXchange Consortium via the PRIDE partner

repository under accession code PXD060138. Further Source data are provided with this paper.

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

## Acknowledgements
We thank D. Bruns, M. Ryan, P. Niewiadomski, D. Wachten, J. Mansfeld, L. Prates Roma, J. Riemer and J. Copeland for reagents, D. Jann, S. Plant, E. Schuster, V. Andre and M. Lerner for experimental assistance, D.K. Breslow for helpful discussions and comments on the manuscript. We thank all members of the Mick lab for stimulating discussions. This work was supported by Deutsche Forschungsgemeinschaft (DFG) grants INST 256/549-1 and INST 256/508-1, and funding to DUM (TRR152-P28 – Project-ID 239283807, FOR5547-P3 – Project-ID 503306912, and Project-ID 513767027) and to BS (CRC1027-C9 – Project-ID 200049484), respectively. DW and PW were funded by the DFG under the Heisenberg Program and FOR5547 (WA3365/5-1 & WA3365/6-1) and by Germany's Excellence Strategy (CIBSS – EXC 2189 – Project ID 390939983). We are grateful to F. Stein at the EMBL proteomics core facility for MS data analysis.

## Author contributions
Conceptualization, T.J.S. and D.U.M.; Methodology, T.J.S., L.K.S., A.P., P.H., J.H., V.T., D.W., D.Y., K.v.d.M., V.C., B.S., and K.F.; Formal analysis, T.J.S., L.K.S., A.P., K.v.d.M., V.C., B.S. and K.F.; Writing – original draft, T.J.S. and D.U.M.; Writing – review & editing, T.J.S., K.F., K.v.d.M., A.P., J.H., B.S. and D.U.M.; Visualization, T.J.S., L.K.S., K.F., A.P., V.C. and B.S.; Funding acquisition, D.U.M.; Supervision, K.F., P.W. and D.U.M.

## Funding

## Competing interests
The authors declare no competing interests.
