## [Transparent Peer Review file · Nature Communications]

An enzymatic cascade enables sensitive and specific proximity labeling proteomics in challenging biological systems

Corresponding Author: Professor David Mick

Version 0:

Reviewer comments:

Reviewer #1

(Remarks to the Author)

Proximity labeling approaches use various promiscuous enzymes to probe protein interactions in vivo. Over the years, these systems have been refined to increase their temporal and spatial resolution. APEX-based proximity labeling offers the highest temporal resolution among these methods. However, its limitations include toxicity due to the conditions required for biotin activation, restricted applicability in certain cell lines, and a high background signal. This manuscript by Sroka et. al. introduces iAPEX, an improved version of the APEX proximity labeling technique, designed to minimize hydrogen peroxide toxicity and reduce non-specific background labeling. This approach employs the enzyme D-amino acid oxidase (DAAO) from *Rhodotorula gracilis*28 to locally generate H₂O₂.

The development of iAPEX addresses key limitations of the original APEX method, particularly cytotoxicity and non-specific labeling. By refining these aspects, iAPEX has the potential to improve the accuracy of proximity labeling experiments, thereby advancing our understanding of subcellular protein interactions. Specifically, the authors applied this approach to the primary cilium, a crucial signaling organelle, which led to identification of new ciliary proteins and quantitative comparative analysis of ciliary proteomes of different cell lines. This highlights the method's applicability to organelles that pose inherent challenges for proteomic studies due to their semi-closed nature, dynamic interactions, and small size. The new approach developed by the authors uses a versatile workflow that enhances the specificity and efficiency of subcellular proteomics studies.

To sum up, this study is valid and timely, especially for analyzing subcellular compartments, in this study primary cilia. It is very well-conducted as well. I recommend that this article be published in Nature Communications with minor revisions. My comments are outlined below:

1. Is iAPEX applicable to retinal pigment epithelial (RPE1) cells, which are very commonly used in studies focusing on the primary cilium. Do iAPEX and DAAO constructs localize to the cilium in RPE1 cells?
2. The authors validated some uncharacterized hits in IMCD3 cells through localization experiments. Did they also validate hits from 3T3 cells? Given that the ciliary proteome overlap between these two cell lines is only ~50%, it would be important to confirm some 3T3-specific candidates in this cell line, which would strengthen their conclusions on cell type-specific differences of the cilium proteome.
3. To demonstrate the broad applicability of the iAPEX approach, could the authors show how it addresses APEX-related challenges in organelles with other inherent complexities, such as mitochondria or similar structures?
4. The authors assessed oxygen consumption in cilia-iAPEX IMCD3 cells to evaluate the system's impact on oxidative stress. However, they did not investigate the specific types of oxidative stress induced by this system in an in vivo setting (Figure 6D–L). Additionally, the use of D-Met in *Xenopus laevis* embryos requires further discussion in terms of embryos' metabolism.
5. How did the authors determine the cutoff value of 23 for TMT enrichment ratios? What criteria were used for this selection? Did they examine known ciliary proteins within the dataset to establish this threshold?
6. I recommend authors to explain the rationale for why they choose D-amino acid oxidase (DAAO) from *Rhodotorula gracilis*.
7. The quantifications in the graphs, such as in Figure 2C and 2D, lack statistical information. The authors should include statistical analyses.

8. In Figure S3A, the control is labeled as IFT88^{-/-}. However, in the text and main figure, it is referred to as CEP164^{-/-}. Which one is the no cilia control? Please clarify.

Reviewer #2

(Remarks to the Author)

Sroka et al. in this manuscript describe a proof-of-concept of their significant improvement of the existing cilia-APEX2 proximity proteomics approach to identify the proteome of an organelle, in this case the primary cilium, that cannot be efficiently purified by classical fractionation methods. This existing approach requires exogenous H₂O₂ supplementation to activate cilia-delivered ascorbate peroxidase (APEX) and induce the protein biotinylation process, which has toxicity and specificity issues that reduce its versatility.

To optimize the biotinylation process, they have designed an elegant way for local (ciliary) enzymatic production of H₂O₂ by delivering D-amino acid oxidase (DAAO) also locally, using the same ciliary localization sequence (a fragment of the ciliary NPHP3 protein) as ascorbate peroxidase (APEX). Instead of exogenous supplementation of H₂O₂ and biotin tyramide, they now supplement with the D-amino acids of choice (they show that several are efficiently oxidized to produce H₂O₂, but D-methionine most efficiently) and biotin tyramide. They also demonstrate the specificity issues that the existing exogenous supplementation procedure has in certain cell lines, due to endogenous cellular peroxidases. They show by IF analysis and immunoblots that the optimized versatile workflow, called iAPEX (in situ APEX activation) is perfectly suited to target the primary cilium in cultured cells, increasing the specificity and output and lowering the background. They also show that they circumvent previous issues in specific cell lines, generating a more specific ciliary proteome with higher efficiency. To demonstrate potential in vivo use, they generated a lentiviral vector to encode both cilia-directed enzymes and present localization data of these in *Xenopus* brain.

The manuscript is clearly written and well-structured, providing informative experimental data of the efficient targeting of both enzymes to the primary cilia of different cell lines, the resulting specific biotinylation and efficient purification and subsequent MS analysis. The figures are in general clear and convincing, some exceptions indicated below, and the pros and cons are outlined well in the discussion. The improvement of iAPEX over the existing cilia-APEX2 is clear and convincingly presented.

Some parts of the manuscript however are more complete than others, and some of the concepts are not (yet) demonstrated convincingly, rendering some remaining questions and concerns. Also some minor points of potential improvement were identified, all indicated point by point below.

MAJOR POINTS:

(1) The authors acknowledge the relevant potential issue with toxicity that requires better validation and/or discussion. The used D-amino acid oxidase (DAAO) was used in other studies to induce oxidative stress. The presented oxygen consumption analysis may be a good proxy for potential cellular toxicity, but it does not exclude the local oxidation for proteins due to the targeted delivery and in that way the potential local concentration of H₂O₂ which could locally disrupt proteins. The assumption that the locally produced H₂O₂ would be immediately consumed by nearby APEX2 (line 107) is later shown to be incorrect through expansion microscopy, as the ciliary biotinylation progresses beyond the localization of the enzymes. This should not be left unmentioned. Similarly, the local acidification due to the generation of keto-acids and ammonium may add to protein disruption. The potential toxicity of this combination is mentioned in the discussion (lines 331-334), but only in the overall cellular context and not taking potential local concentrations into account.

(2) The assessment of the proteomics outcome to the other proximity datasets (Fig. 5E) could be better represented and discussed: more datasets have been generated that may provide further insights, therefore a more extensive Venn diagram, with more information regarding the overlapping proteins in the different datasets, is warranted. Important points currently missing are: What was the overlap, for example, between Cilia-APEX2 from May et al, also performed in IMCD3 cells, and the cilia-APEX2 dataset produced in the current study?

(3) Regarding both previous points, the identified proteins enriched in cilia-APEX2 experiments, but not in cilia-iAPEX may not just represent false positives, but could also be ciliary proteins that are disrupted by the local activities of H₂O₂, keto-acids and NH₃. This should be discussed.

(4) Another point left undiscussed are known ciliary proteins potentially "missing" from the dataset. What percentage of ciliopathy-associated proteins are identified? Is there a skewed representation of known ciliary GPCRs, centrosomal proteins, transition zone proteins, membrane proteins, axonemal proteins? Is this due to the NPHP3-derived ciliary localization signal? How does this relate to the Cilia-APEX2 method?

(5) Although the authors have performed substantial experimentation to convincingly prove the specificity and versatility of the developed iAPEX pipeline for primary cilia, broader use is suggested (lines 390-395) but not demonstrated. The proven success of the current method is strongly dependent on the 200aa ciliary localization tag of NPHP3 that they have employed and which works well for biotinylation of the ciliary proteome in cell lines as they have demonstrated. They indicate potential broader use for other subcellular compartments or subpopulations of proteins, but have not shown this experimentally nor do

they mention any particular localization signals that they have in mind. Importantly, they also do not link it to the U-ExM data showing that the biotinylation is not strictly localized to both enzymes, suggesting diffusion of the signal. This is an added benefit in case the organelle is confined and the proteome of the entire organelle is the target, but not if you want to assess a local interaction at molecular distance, as with the BirA-based methods.

(6) The experiment performed for broader in vivo use of iAPEX in *Xenopus* does show some colocalization of the two enzymes, but the resulting cilia-specific biotinylation is much less convincing than in the cell lines as the microscopy shown in FIG. 6 lacks the required resolution. It is logical that the cilia-APEX2 and cilia-DAAO signals overlap well, as they use the same localization tag and are encoded by a single construct, but overlay with acetylated tubulin and biotin signals is often incomplete or even poor. This should at least be indicated and discussed. Also, it remains unclear why this model was not completed with a protein purification and mass spectrometry experiment, as that would have been definite proof-of-concept. If this is due to current limitations that should still be overcome, that should then also be critically discussed as it is one of the presented benefits.

(7) Also, the microscopy data of novel ciliary proteins that were presented (FIG. 5F) currently lack convincing quality to prove the ciliary specificity. For some (PSKH1, FCAP71) the difference subciliary signals don't even seem to align well. This can be easily addressed by using a confocal (with Airyscan) rather than epifluorescence microscope, as they have done in the many other micrographs, use representative markers and indicate them clearly (which ones are ARL13B and which ones are acTub marking the cilium?), use different colors for the different antibodies and provide overlays, and provide more examples as a supplement. Currently, this limits the confidence in the presented data, beyond the method. Commonly, to show relevance to cilia function, siRNA or shRNA-based knockdown analysis is also included. This would be a valuable addition to this experiment.

MINOR POINTS:

(8) The fact however that local delivery of D-amino acid oxidase (DAAO) to generate H₂O₂ has already been used several times in other studies and other contexts in mammalian cells, even in combination with APEX2 (Ref. 23) should be indicated in a clearer way.

(9) It remains unclear why the new method is not named iAPEX2, as the APEX2 enzyme is used, which was an optimization of the APEX enzyme. Not naming it iAPEX2 could cause confusion.

(10) Expansion microscopy (FIG. 2 A and B): show quantification of the strength of the different signals related to the ciliary dimensions, allowing a more accurate measurement/representation of the extended localization of the biotin signal.

(11) Lines 156-157: provide a reference to "recently developed proximity labeling methods"

(12) Line 376: what could be the relevance of identifying putative transcription factors in the cilium?

(13) Lines 377-379 "explain clinical differences... to tackle this larger question in the cilia community": It has already been shown that specialized cilia in different cell types have different proteomes, such as the photoreceptor sensory cilium (e.g. Liu et al., Mol Cell Proteomics 2007, PMID: 17494944). This should be included in this discussion: it is not a new concept.

(14) Line 321: typo: cel0l toxicity

(15) western blotting/western blot should be without capitals (used with capitals throughout the text)

(16) It is not clearly visible what the colors in (B) mean in Figures S4 and S5. Please optimize this, perhaps by enlarging the representative part of the clustering graph and adding representative labels to make it comprehensive.

Reviewer #3

(Remarks to the Author)

This manuscript by Sroka et al. describes a clever adaptation to classical APEX2-based proximity labeling. By co-localizing APEX2 with the D-amino acid oxidase (DAAO) enzyme in a targeted subcellular region, the authors elegantly generate hydrogen peroxide in situ, circumventing the widespread off-target labelling and cytotoxicity usually observed when H₂O₂ is added exogenously. They term this approach "iAPEX," and demonstrate that it greatly improves specificity and still retains the high temporal resolution typical of the APEX method. They further show how iAPEX works in multiple mammalian cell lines that have historically shown high background with classic APEX, and also provide a nice proof-of-concept in *Xenopus laevis*.

I see this as a significant advance for proximity labeling, especially if one aims to analyze fine structures (like primary cilia) in cell lines or tissues that are not well-suited to external H₂O₂ addition. The experimental data presented, immunofluorescence, expansion microscopy, Seahorse assays, plus TMT-based proteomics, gives strong support to the claims. The manuscript is clearly structured, well-illustrated, and of broad interest to cell biologists and proteomic researchers. Nevertheless, below I have some suggestions that should further strengthen the final version.

Major comments

1. Kinetic range and concentration considerations

The authors show a thorough characterization with different D-amino acids (D-Met, D-Ala, etc). It is good if they highlight explicitly how widely this approach can be adapted in other cell contexts. Possibly, in some cell types, the amino acid transport or local oxygen availability might limit the H₂O₂ generation. A bit more discussion here would be helpful so that reader's from other fields are not uncertain how best to adapt it.

2. Comparison to related proximity-labeling methods

The authors nicely mention that iAPEX addresses the major limitation of classic APEX. They also reference other new strategies, such as split-APEX, TurboID, or TransitID. However, short paragraph in the Discussion that positions iAPEX in direct comparison (in terms of timescales, background labelling, substrate costs, etc.) would be beneficial for us specialists.

3. In vivo feasibility

The demonstration of iAPEX in *Xenopus laevis* is nice and quite convincing. It would be interesting if the authors expand slightly on potential complexities in adult tissues or in organs with lower oxygen tension, or mention if an optimized DAAO variant might be needed in future.

4. Replicate variation and proteome-scale reproducibility

While the authors do show replicates, it might reassure readers to see more detail on replicates correlations (like Pearson's *r*) or a mention how stable is the identification of cilia proteins across replicates. Even short clarifications or a reference to the cluster correlation in the supplement would be nice for completeness.

5. Novel cilia candidates

Some of the newly identified cilia-localized proteins (for instance, PSKH1 or FCAP71/33) might open new directions in cilia biology. A brief speculation on how they could function or what domain features they carry may be interesting for the cilia community.

Minor comments

- Some phrasing could be toned down or rearranged for conciseness in the main figures, as they contain multiple panels. Also, clarifying the color-coding (like green for APEX2, red for biotin, etc.) more explicitly in the figure legends might help.
- For the Seahorse-based measurements, it would be helpful to mention approximate cell densities in the figure legends, so readers can replicate the oxygen consumption assays more precisely.
- In the text, "non-specific labeling" vs "off-target labeling" are used somewhat interchangeably. Perhaps unify the terminology in the final revision.

Reviewer #4

(Remarks to the Author)

The manuscript by Sroka et al. presents a groundbreaking advancement in proximity labeling technology based on APEX2. This technique was originally developed by Alice Ting's group at MIT, Boston, for mitochondria and other organelles. The authors have now successfully adapted it to primary cilia. One of the long-standing limitations of the classical APEX2 approach has been the specificity of labeling—partly due to potential mislocalization of the APEX enzyme, which in the traditional method is activated by externally added biotin-phenol and hydrogen peroxide (H₂O₂).

This manuscript addresses this limitation in a highly elegant manner by eliminating the need for exogenous H₂O₂ altogether. Instead, the authors employ local production of H₂O₂ through the targeted expression of a D-amino acid oxidase (DAAO) within the cilium itself. They refer to this refined approach as *in situ* APEX (iAPEX). This clever strategy increases labeling specificity, as both the APEX-tagged target protein and the H₂O₂-generating DAAO must be present at the same subcellular site—here, the cilium—for effective biotinylation to occur. And it allows to use APEX2 in cells that were not accessible to the method before due to massive background peroxidase activity.

Cilia pose a particular challenge in many respects: each cell typically forms only a single primary cilium, making the available material extremely limited. At the same time, primary cilia are at the center of a rapidly growing field of research, particularly in the context of ciliopathies—a group of genetic disorders caused by defects in ciliary structure or function, for which therapeutic options are still largely lacking.

From my perspective, this manuscript provides an elegant and innovative solution that allows for a more refined analysis of cilia and ciliopathies. I find the approach highly compelling, especially given its relevance to an important and currently under-addressed medical field. Preliminary data were already presented at the 2024 Cilia Meeting in Dublin, where they generated substantial excitement, and are available as preprint.

That said, there are several points that the authors should address:

1. Temporal Resolution: The manuscript states a 5-minute labeling time (line 156), which contrasts with the commonly used 1-minute H₂O₂ pulse applied in previous studies, including those by Mick et al. A more nuanced discussion of the temporal resolution would be appreciated. I would also recommend softening the language in the abstract, where the phrase "live cell proteomics" may be somewhat overstated.
2. Validation of Ciliary Localization: The validations using the ALFA tag appear relatively weak. To convincingly demonstrate the specificity of the anti-ALFA staining, negative controls should be included.
3. In Vivo Application: It would be exciting to see whether iAPEX can be applied *in vivo*. Is there a specific reason why biotin-phenol/D-Norvaline treatment was performed post-fixation? Clarification and discussion of this point would be helpful.
4. Statistical Analysis: Statistical analyses should be included for the data presented in Figure 2.
5. To better interpret the findings presented in Figure 1a, additional imaging parameters and methodological details would be helpful.
6. The authors report APEX2 and DAAO localization at the ciliary membrane in Fig.1, even though NPHP3 n-term was used

as the targeting motif. Could this reflect an artifact related to expansion microscopy? Co-staining with established ciliary membrane markers, in addition to acetylated tubulin, could help clarify this point.

Version 1:

Reviewer comments:

Reviewer #1

(Remarks to the Author)

The authors have satisfactorily addressed all of my comments through new experiments and/or reanalysis of their data. Their responses were convincing, and the revisions have strengthened the manuscript.

I have also reviewed their rebuttal to the comments raised by the other reviewers. The authors have provided clear and thorough explanations and included data from new experiments, which in my opinion convincingly resolve the issues raised.

Overall, the revised version represents a significant improvement over the original submission. I recommend the manuscript for publication in its current form.

Reviewer #2

(Remarks to the Author)

Sroka et al. have provided an informative and comprehensive rebuttal, responding to all of my comments in a comprehensive and clear manner. They have robustly addressed the identified shortcomings and questions without raising new ones where possible, and added experimental data where needed and where relevant in the context of this paper. I have no remaining questions and consider the current revised manuscript an excellent contribution to Nature Communications, of potential interest to a broad readership.

Reviewer #3

(Remarks to the Author)

The authors have done good job on addressing the criticism raised and in doing so improved the manuscript.

Reviewer #4

(Remarks to the Author)

All the points of criticism I previously raised were fully addressed by the authors in the revised manuscript. From my perspective, there are no objections that would speak against the publication of this exciting paper!

We thank all the reviewers for their efforts to help improve our manuscript and have attached a point-by-point response to all reviewers' comments below (in blue).

REVIEWER COMMENTS

Reviewer #1 (Remarks to the Author):

Proximity labeling approaches use various promiscuous enzymes to probe protein interactions in vivo. Over the years, these systems have been refined to increase their temporal and spatial resolution. APEX-based proximity labeling offers the highest temporal resolution among these methods. However, its limitations include toxicity due to the conditions required for biotin activation, restricted applicability in certain cell lines, and a high background signal. This manuscript by Sroka *et al.* introduces iAPEX, an improved version of the APEX proximity labeling technique, designed to minimize hydrogen peroxide toxicity and reduce non-specific background labeling. This approach employs the enzyme D-amino acid oxidase (DAAO) from *Rhodotorula gracilis*28 to locally generate H₂O₂.

The development of iAPEX addresses key limitations of the original APEX method, particularly cytotoxicity and non-specific labeling. By refining these aspects, iAPEX has the potential to improve the accuracy of proximity labeling experiments, thereby advancing our understanding of subcellular protein interactions. Specifically, the authors applied this approach to the primary cilium, a crucial signaling organelle, which led to identification of new ciliary proteins and quantitative comparative analysis of ciliary proteomes of different cell lines. This highlights the method's applicability to organelles that pose inherent challenges for proteomic studies due to their semi-closed nature, dynamic interactions, and small size. The new approach developed by the authors uses a versatile workflow that enhances the specificity and efficiency of subcellular proteomics studies.

To sum up, this study is valid and timely, especially for analyzing subcellular compartments, in this study primary cilia. It is very well-conducted as well. I recommend that this article be published in Nature Communications with minor revisions. My comments are outlined below:

We thank the reviewer for this positive feedback and appreciate the suggestions to strengthen the manuscript, which we are addressing in a point-by-point response below.

1. Is iAPEX applicable to retinal pigment epithelial (RPE1) cells, which are very commonly used in studies focusing on the primary cilium. Do iAPEX and DAAO constructs localize to the cilium in RPE1 cells?

Yes, the technology has so far been applicable to all cell lines tested, including RPE-1 cells. We have included data to demonstrate the proper localization of the transgenes and biotinylation after activation in the new Fig. S7B.

2. The authors validated some uncharacterized hits in IMCD3 cells through localization experiments. Did they also validate hits from 3T3 cells? Given that the ciliary proteome overlap between these two cell

lines is only ~50%, it would be important to confirm some 3T3-specific candidates in this cell line, which would strengthen their conclusions on cell type-specific differences of the cilium proteome.

After these suggestions we have cloned several ciliary candidate proteins identified in our iAPEX proteomics dataset from NIH/3T3 cells, and could confirm the presence of FHDC1 as well as CKAP5 in NIH/3T3 primary cilia (**new Fig. 5G**). The presence of the formin FHDC1 in primary cilia of fibroblasts was reported in literature (Copeland et al., 2018; PMID: 29742020). FHDC1 overexpression led to lengthening of primary cilia, which we could also confirm (see **new Fig. 5G**). CKAP5/XMAP215 is a TOG-domain containing protein and microtubule polymerizer known to bind to MT tips (Brouhard et al., 2008; PMID: 18191222). This data adds more microtubule-associated proteins (MAPs) to the primary cilia proteome of NIH-3T3 cells. After TOGARAM, CKAP5 is the second TOG-domain containing MAP in primary cilia. While we identified TOGARAM in both IMCD3 and NIH-3T3 cells (similar to the non-TOG MAP, CKAP2), CKAP5 appears to be a 3T3-specific ciliary MAP. CKAP2L seems IMCD3-specific. We have added a new section to the discussion about potential functions of selected newly identified proteins (see response to reviewer #3). Unfortunately, we could not find working antibodies against these proteins to confirm their specificity for either cell type.

3. To demonstrate the broad applicability of the iAPEX approach, could the authors show how it addresses APEX-related challenges in organelles with other inherent complexities, such as mitochondria or similar structures?

We have generated new constructs and cell lines to address whether the technology also works in cellular contexts other than primary cilia. We have included a **new Fig. 6** to show that the system in principle also works in the context of two other -very different- organelles, namely mitochondria and lipid droplets (LDs). In mitochondria, a major challenge is that substrates must cross both the outer and inner mitochondrial membranes to reach the localized enzymes, which is not trivial for amino acids (see response to reviewer #3). Despite this limitation, we demonstrate that compartment-specific iAPEX labeling can be achieved, as shown in the **new Fig. 6B**.

The challenges in LDs are two-fold (see **new Fig. 6C**): 1) their membrane curvature is opposite of the primary cilia membrane, such that it does not enclose an aqueous compartment but is open to the cytoplasm; 2) many LD-associated proteins are present not only on the LD surface but also in other compartments, such as the cytosol or the endoplasmic reticulum (ER). By directing DAAO and APEX2 to LDs by two independent fusion proteins, namely UBXD8-APEX2 and DAAO-PLIN5, we can show biotinylation specifically on the LD surface where the two enzymes co-localize when we optimize the labeling conditions (see **new Fig. 6D**; also see comment 5 by reviewer #2). This new data also highlights the specificity of the iAPEX approach over conventional APEX2 labeling.

4. The authors assessed oxygen consumption in cilia-iAPEX IMCD3 cells to evaluate the system's impact on oxidative stress. However, they did not investigate the specific types of oxidative stress induced by this system in an in vivo setting (Figure 6D–L). Additionally, the use of D-Met in *Xenopus laevis* embryos requires further discussion in terms of embryos' metabolism.

We used the oxygen consumption measurements to assess the amounts of H₂O₂ produced, according to published procedure. We did not investigate other forms of oxidative stress, as the enzyme is known to induce a specific and well-characterized oxidative stress, namely H₂O₂ production (Alim et al., 2014; Smolyarova et al., 2022). Yet, we found that injection of mRNA encoding cilia-iAPEX into four-cell *Xenopus* embryos followed by administration of D-amino acids - either by incubation of embryos in a) 10 mM D-methionine / D-norvaline from st. 40 - 45 (~ 2 days) or b) 25 mM D-methionine / D-norvaline for up to 4 h at st. 45 or by intraperitoneal injection of ~13 nmol D-norvaline into st. 45 tadpoles and incubation for up to 6 h - does not lead to apparent developmental defects until st.45 (we would be happy to provide

such data if deemed useful for the reader). This contrasts with TurboID applications, which have been reported to consume biotin and therefore lead to developmental defects in *Drosophila melanogaster* (Branon et al., 2018). As our established procedure for the in vivo iAPEX labeling is based on ventricular injections of D-AAAs with much shorter incubation times (up to 30 min), the amount of oxidative damage accumulated over longer periods of time is unclear.

How exactly *Xenopus laevis* embryos metabolize D-amino acids, including D-Met is not fully resolved. However, we have switched our efforts to using the non-proteinogenic D-amino acid, D-norvaline (D-Nva) in the **new Figures 7K-L** and **S7C-G**. L-Nva is a known inhibitor of urea synthesis in isolated rat liver cells (Rognstad, 1977), and also inhibits arginase in vivo due to its structural similarity with ornithine (Saheki et al., 1979). However, these effects are stereoisomer-dependent. Importantly, D-Nva has been reported to have no effect on neuronal function (Kalinichenko et al., 2023), therefore we resorted to D-Nva for our *Xenopus* brain experiments. Data on the antioxidant metabolism of developing *Xenopus laevis* indicated low protection from oxidative damage during the first 3 days of development until st.40/41 (Rizzo et al., 2007), after which the expression of enzymes to protect from oxidative damage is increased. Most experiments in our study were performed at st.45, when oxidative damage induced by DAAO catalysis of D-Met or D-Nva should be low. Nonetheless, we have expanded the discussion on amino acid metabolism and potential effects by the local change in pH (see response to reviewer #2).

5. How did the authors determine the cutoff value of 23 for TMT enrichment ratios? What criteria were used for this selection? Did they examine known ciliary proteins within the dataset to establish this threshold?

Indeed, the cutoff value of 2^3 for the TMT enrichment ratios in the initial analysis was guided by the presence of known cilia proteins. This empiric method is currently common procedure; however, it is prone to bias. As we show in **Fig. S3**, this somewhat arbitrary selection of enrichment ratios is inferior to hierarchical cluster analyses for identifying cilia proteins, which we performed and compared directly to the cutoff strategy.

6. I recommend authors to explain the rationale for why they choose D-amino acid oxidase (DAAO) from *Rhodotorula gracilis*.

Thank you for this suggestion, we have added a rationale (with citations) for why we chose the DAAO enzyme. It is a very well characterized and established enzyme, with many quantitative biotechnology as well as in vivo applications.

7. The quantifications in the graphs, such as in Figure 2C and 2D, lack statistical information. The authors should include statistical analyses.

This was a valuable remark, and we added statistical information to these graphs (as well as to **Fig. S2** and the **new Fig. S7A**) to demonstrate significant increases in local biotinylation.

8. In Figure S3A, the control is labeled as IFT88^{-/-}. However, in the text and main figure, it is referred to as CEP164^{-/-}. Which one is the no cilia control? Please clarify.

Thanks for pointing out that this was unclear. The no cilia control is indeed the CEP164^{-/-} cell line (lanes 2, 5 and 6 in the previous figure, now lanes 1, 4 and 5). However, in the previous lane 1, we additionally loaded cell lysate from an IFT88^{-/-} cell line, as the anti-IFT88 antibody batch (PTG #13967-1-AP) we

used in **Fig. 4B** was of lower quality. To avoid confusion, we have decided to cut out lane 1 from **Fig. S3A**, however, decided to keep it in **Figs. 4B, 5B** and **5C**, where it serves as a technical control. Lane numbering in **Fig. S3A** has been adjusted accordingly.

Reviewer #2 (Remarks to the Author):

Sroka et al. in this manuscript describe a proof-of-concept of their significant improvement of the existing cilia-APEX2 proximity proteomics approach to identify the proteome of an organelle, in this case the primary cilium, that cannot be efficiently purified by classical fractionation methods. This existing approach requires exogenous H₂O₂ supplementation to activate cilia-delivered ascorbate peroxidase (APEX) and induce the protein biotinylation process, which has toxicity and specificity issues that reduce its versatility.

To optimize the biotinylation process, they have designed an elegant way for local (ciliary) enzymatic production of H₂O₂ by delivering D-amino acid oxidase (DAAO) also locally, using the same ciliary localization sequence (a fragment of the ciliary NPHP3 protein) as ascorbate peroxidase (APEX). Instead of exogenous supplementation of H₂O₂ and biotin tyramide, they now supplement with the D-amino acids of choice (they show that several are efficiently oxidized to produce H₂O₂, but D-methionine most efficiently) and biotin tyramide. They also demonstrate the specificity issues that the existing exogenous supplementation procedure has in certain cell lines, due to endogenous cellular peroxidases. They show by IF analysis and immunoblots that the optimized versatile workflow, called iAPEX (in situ APEX activation) is perfectly suited to target the primary cilium in cultured cells, increasing the specificity and output and lowering the background. They also show that they circumvent previous issues in specific cell lines, generating a more specific ciliary proteome with higher efficiency. To demonstrate potential in vivo use, they generated a lentiviral vector to encode both cilia-directed enzymes and present localization data of these in *Xenopus* brain.

The manuscript is clearly written and well-structured, providing informative experimental data of the efficient targeting of both enzymes to the primary cilia of different cell lines, the resulting specific biotinylation and efficient purification and subsequent MS analysis. The figures are in general clear and convincing, some exceptions indicated below, and the pros and cons are outlined well in the discussion. The improvement of iAPEX over the existing cilia-APEX2 is clear and convincingly presented.

We thank the reviewer for this overall positive assessment of our new technology.

Some parts of the manuscript however are more complete than others, and some of the concepts are not (yet) demonstrated convincingly, rendering some remaining questions and concerns. Also some minor points of potential improvement were identified, all indicated point by point below.

MAJOR POINTS:

(1) The authors acknowledge the relevant potential issue with toxicity that requires better validation and/or discussion. The used D-amino acid oxidase (DAAO) was used in other studies to induce oxidative stress. The presented oxygen consumption analysis may be a good proxy for potential cellular toxicity, but it does not exclude the local oxidation for proteins due to the targeted delivery and in that way the potential local concentration of H₂O₂ which could locally disrupt proteins. The assumption that the locally

produced H₂O₂ would be immediately consumed by nearby APEX2 (line 107) is later shown to be incorrect through expansion microscopy, as the ciliary biotinylation progresses beyond the localization of the enzymes. This should not be left unmentioned. Similarly, the local acidification due to the generation of keto-acids and ammonium may add to protein disruption. The potential toxicity of this combination is mentioned in the discussion (lines 331-334), but only in the overall cellular context and not taking potential local concentrations into account.

We thank the reviewer for this important comment, which gives us the chance to clarify our statements. As the reviewer pointed out, it appears likely from our data that H₂O₂ is not fully consumed by nearby APEX2 enzymes. We therefore re-phrased “consumed” to “used”. In fact, how much H₂O₂ is consumed/used by APEX2 is unclear. We interpret our findings that H₂O₂ is indeed used at the location of the APEX2 enzymes, but we also find it very likely that excess H₂O₂ generated at the location of the DAAO enzymes diffuses further out into the primary cilium lumen, causing potential local disruptions (see below). We consider two potential (not mutually exclusive) explanations for why we observe biotinylation not restricted to the location of the APEX2 enzyme: 1) biotinylated proteins in close proximity to the enzyme may well diffuse/or are actively transported away from the enzyme, 2) the APEX2 enzyme may generate excess phenoxyl radicals that saturate local reaction sites and diffuse into the periphery. New data presented in **Fig. 6**, where we titrate DAAO substrates to spatially restrict labeling in two applications supports the H₂O₂-dependent labeling distance.

Indeed, the overall toxicity/damage for the cell appears to be minor, as cells continue growing normally after 30 min of iAPEX labeling and *Xenopus laevis* embryos do not show any obvious developmental delay when injected with enzymes and substrates (see response to reviewer #1). To assess local toxicity of the iAPEX system, we now provide new data presented in **Videos 1-3**. Here, we assessed the intraflagellar transport (IFT), a central primary cilia mechanism for protein transport. After 30 min of local H₂O₂ production (**Video 2**) or iAPEX labeling (**Video 3**) IFT appeared unaffected compared to no treatment (**Video 1**). While we cannot rule out local disruptions of other processes, we believe this will only have a minor impact on the technological application of iAPEX, as the labeling is performed for a relatively short amount of time (5 – 30 min), right before fixation or lysis of the specimen.

Yet, in the timeframe of the iAPEX labeling potential redox signaling effects may occur, which we account for in important technical controls by activating DAAO while omitting biotin tyramide (+D-Met –DTBT) in all proteomics experiments.

We have now added potential local disruptions into the discussion (see response to reviewer #1).

(2) The assessment of the proteomics outcome to the other proximity datasets (Fig. 5E) could be better represented and discussed: more datasets have been generated that may provide further insights, therefore a more extensive Venn diagram, with more information regarding the overlapping proteins in the different datasets, is warranted. Important points currently missing are: What was the overlap, for example, between Cilia-APEX2 from May et al, also performed in IMCD3 cells, and the cilia-APEX2 dataset produced in the current study?

This is an important point raised by the reviewer. Indeed, and we are happy to expand on the comparison to other datasets. The previous version of this manuscript already contained a comparison of the cilia-iAPEX datasets from this study with the cilia-APEX2 dataset from May et al. (**Fig. 5E**). In addition, we have performed additional analyses, which we present in new Venn diagrams comparing our datasets to the other available technologies, i.e. the TurboID-based dataset from Liu et al., 2024 (see **new Figs. S6A-B**). Although also performed in IMCD3 cells, we decided not to add a comparison to the BioID2 dataset from Aslanyan et al., 2023, as they in fact fused an ubiquitin binding domain to their transgene, which is likely to result in alterations in cilia, as it may recruit or hold back ubiquitinated proteins, such that we believe a direct comparison is flawed. As suggested, we also added a comparison between the cilia-APEX2 proteome from May et al. to our new cilia-APEX2 dataset (**new Fig. S6D**) and also compared it in respect to ciliopathy genes (see also response to comment #4). It should be noted, however, that the

May et al. cilia proteome has been compiled from multiple datasets, while here we only generated one dataset each for IMCD3 cells. Hence, higher sensitivity in May et al. was to be expected. Nonetheless, in the new version of the manuscript we provide the reader with more information and now compare our findings to other datasets.

(3) Regarding both previous points, the identified proteins enriched in cilia-APEX2 experiments, but not in cilia-iAPEX may not just represent false positives, but could also be ciliary proteins that are disrupted by the local activities of H₂O₂, keto-acids and NH₃. This should be discussed.

We thank the reviewer for this comment. As pointed out in comment #1, we cannot fully rule out potential damage to ciliary proteins, which may be degraded by a quality control system. While we envision that changes in pH are more relevant for protein folding, oxidation of proteins/peptides might have more severe consequences for peptide identification and quantitation by mass spectrometry. In particular, when using TMT-based quantitation, absence of a specific TMT-label may not be due to absence of a protein but oxidative damage resulting in a mass shift. We therefore searched our mass spectrometry data for oxidized peptides and found that the distribution of oxidized and non-oxidized peptides across samples, independent of the iAPEX labeling reaction, is highly similar (**new Fig. S4D**). This is true both for known cilia proteins (exemplified by IFT proteins) as well as for the proteins in the potential false-positive clusters 6 & 7 (see **Fig. 4F**). As suggested by the reviewer we have added this discussion to the manuscript.

(4) Another point left undiscussed are known ciliary proteins potentially “missing” from the dataset. What percentage of ciliopathy-associated proteins are identified? Is there a skewed representation of known ciliary GPCRs, centrosomal proteins, transition zone proteins, membrane proteins, axonemal proteins? Is this due to the NPHP3-derived ciliary localization signal? How does this relate to the Cilia-APEX2 method?

Indeed, a more comprehensive comparison was missing in the previous version of the manuscript. As mentioned in a previous comment (#3), we have now included information on the coverage of ciliopathy-associated proteins (**new Figs. S6B-C**), of which we cover roughly one third (according to Turan et al., 2023). Considering that many proteins required for cilium formation are not physically located in primary cilia and that there will likely be cell type-specific components (e.g. ciliary GPCRs), it is unclear what fraction could be expected.

In our revised manuscript we also briefly discuss the apparently skewed representation of axonemal (“overrepresented”) and membrane (“underrepresented”) proteins in the iAPEX versus previous cilia-APEX2 datasets. However, we want to express our opinion that there appears not enough high-quality data to fully interpret these findings. The apparent reduction in certain proteins could be due to 1) “missing” proteins due to low sensitivity (membrane proteins are generally harder to detect by mass spectrometry), or 2) higher specificity of labeling (transition zone or centrosomal components may no longer be labeled due to better spatial resolution). Therefore, we find it difficult to differentiate what the exact reason for this skewed representation may be, as many explanations appear possible in addition to the protein’s properties, such as size, surface-exposed tyrosine residues (acceptor for phenoxy radicals) etc.

Nonetheless, we have added more information by also providing protein name lists in our **new Figs. S6A-C**.

(5) Although the authors have performed substantial experimentation to convincingly prove the specificity and versatility of the developed iAPEX pipeline for primary cilia, broader use is suggested (lines 390-395) but not demonstrated. The proven success of the current method is strongly dependent on the 200aa

ciliary localization tag of NPHP3 that they have employed and which works well for biotinylation of the ciliary proteome in cell lines as they have demonstrated. They indicate potential broader use for other subcellular compartments or subpopulations of proteins, but have not shown this experimentally nor do they mention any particular localization signals that they have in mind. Importantly, they also do not link it to the U-ExM data showing that the biotinylation is not strictly localized to both enzymes, suggesting diffusion of the signal. This is an added benefit in case the organelle is confined and the proteome of the entire organelle is the target, but not if you want to assess a local interaction at molecular distance, as with the BirA-based methods.

We thank the reviewer for raising this point. We also hope that the reviewer agrees that for the purpose of organellar proteomics that we have applied the system for, generating larger amounts of phenoxyl radicals in a short amount of time will be advantageous for acquiring high temporal resolution. Nevertheless, as we also agree with the reviewer that many proximity labeling applications may require higher spatial resolution of labeling in other cellular contexts. Therefore, we have added new experimental data and expanded our manuscript by three additional applications (see **new Fig. 6**; see response to comment 3 raised by reviewer #1):

1) We have included data that shows spatially restricted labeling in primary cilia. Fusing APEX2 to the transcription factor GLI2 (GFP-APEX2-GLI2) directs the APEX2 enzyme specifically to the ciliary tip (see **Fig. 6A** and **Fig. S7A**). By combining this construct with cilia-DAAO, we can show that biotinylation in primary cilia can be restricted to the tip by lowering the D-amino acid concentrations, which results in less H₂O₂ production by cilia-DAAO.

2) We have added an additional application of the iAPEX technology to study lipid droplets, which provides a different cellular context and also addresses the localization of labeling, which was restricted to the LD surface when optimizing the labeling condition by lowering D-amino acid concentration and labeling times (**new Figs. 6C,D**).

3) We have added another non-cilia application by showing that iAPEX labeling is also possible in the context of mitochondria (**new Fig. 6B**).

While we agree with the reviewer that currently the BirA-based methods are the go-to methods when looking at local interactions at molecular distance, these experiments show that the iAPEX system also offers an alternative method when restricting H₂O₂ production by optimizing labeling times and substrate concentrations.

(6) The experiment performed for broader in vivo use of iAPEX in *Xenopus* does show some colocalization of the two enzymes, but the resulting cilia-specific biotinylation is much less convincing than in the cell lines as the microscopy shown in FIG. 6 lacks the required resolution. It is logical that the cilia-APEX2 and cilia-DAAO signals overlap well, as they use the same localization tag and are encoded by a single construct, but overlay with acetylated tubulin and biotin signals is often incomplete or even poor. This should at least be indicated and discussed. Also, it remains unclear why this model was not completed with a protein purification and mass spectrometry experiment, as that would have been definite proof-of-concept. If this is due to current limitations that should still be overcome, that should then also be critically discussed as it is one of the presented benefits.

We have added new data with improved imaging resolution. However, the experimental approach is challenging for several reasons:

1) since this is a proof-of-concept study, we do not have transgenic lines yet, but mRNA has to be injected into each embryo, which is then reared for five days and injected with labeling solution into the ventricular system. The different degrees of overlap between the enzymes and acetylated α -Tubulin in some cells is likely because not all cells will receive the same amount of mRNA. Similarly, we often observe an incomplete overlay between the enzymes and biotin signal (also in tissue culture, see **Fig. 3A, S7B**, or

the ExM images in **Figs. 2A-B**). This we interpret as possible transport of labeled proteins or diffusion of phenoxy radicals away from the enzyme (see response to comment 1).

2) Tissues must be dissected and stained ex situ, and imaging is then performed on the mounted tissue at mm-scale. The tissue has an extensive 3D structure which makes imaging challenging: the multiciliated ventricle roof epithelium is extremely corrugated and ventral monocilia are squeezed in between the lateral walls of the ventricle. In general, imaging deeper layers of this tissue is much more challenging than 2D cell culture. However, demonstrating here that proximity labeling of these brain cilia is possible despite these challenges opens up the possibility of studying the cilia proteome even in tissues that are hard to access -and in an extremely important developing organ as the brain.

We have also added a new section on limitations for in vivo applications into the discussion. In brief, currently, the amount of material obtained by single *Xenopus* embryos is limited and we would probably require several hundreds. As we used mRNA injections to express the iAPEX transgenes in the current study, this was not feasible, so we will have to generate transgenic animals as a next step.

(7) Also, the microscopy data of novel ciliary proteins that were presented (FIG. 5F) currently lack convincing quality to prove the ciliary specificity. For some (PSKH1, FCAP71) the difference subciliary signals don't even seem to align well. This can be easily addressed by using a confocal (with Airyscan) rather than epifluorescence microscope, as they have done in the many other micrographs, use representative markers and indicate them clearly (which ones are ARL13B and which ones are acTub marking the cilium?), use different colors for the different antibodies and provide overlays, and provide more examples as a supplement. Currently, this limits the confidence in the presented data, beyond the method. Commonly, to show relevance to cilia function, siRNA or shRNA-based knockdown analysis is also included. This would be a valuable addition to this experiment.

We thank the reviewer for his suggestions to improve the data on the newly identified ciliary proteins (see also response to reviewer #4). To address this, we have generated stable cell lines expressing the candidates, have improved imaging and re-worked all the images, which are now presented in **Fig. 5F** and the **new Fig. 5G**. We have also more clearly indicated the used markers and added overlay images with colors indicated.

In order to test the relevance of some of the candidates, we have indeed resorted to siRNA-based knockdown experiments. However, so far, we could only observe minor effects on Hedgehog signaling, which is to our opinion the best, commonly used readout of cilia functions, but does not cover all functions. Yet, as knockdowns have only been partial and we do not want to present lack of evidence for a potential function as evidence for a lack of function, we have decided not to include this data into the current version of the manuscript.

MINOR POINTS:

(8) The fact however that local delivery of D-amino acid oxidase (DAAO) to generate H₂O₂ has already been used several times in other studies and other contexts in mammalian cells, even in combination with APEX2 (Ref. 23) should be indicated in a clearer way.

We hope we have clarified that the frequent use of DAAO to generate H₂O₂ in literature was what sparked us to use DAAO in this context (see response to reviewer #1).

(9) It remains unclear why the new method is not named iAPEX2, as the APEX2 enzyme is used, which was an optimization of the APEX enzyme. Not naming it iAPEX2 could cause confusion.

Although this is a fair point, we picked the name “iAPEX” because in principle the DAAO enzyme could also be combined with the APEX enzyme, which would also lead to “in situ APEX activation”. We are successfully using this combination now in other projects. As we have presented the technology as “iAPEX” at different international conferences (see comments by reviewer #4) and people are referring to this method based on our preprint, we are afraid that renaming it to “iAPEX2” could now create additional confusion.

(10) Expansion microscopy (FIG. 2 A and B): show quantification of the strength of the different signals related to the ciliary dimensions, allowing a more accurate measurement/representation of the extended localization of the biotin signal.

We have added line plots for the data presented in **Figs. 2A-B**, which are presented as new **Figs. S2A-B**. These plots indicate that the localization of the enzymes is indeed restricted to the membrane and not the lumen of cilia (marked by acTub staining, see **Fig. S2A**), while the biotin signals co-localize with the entire cilium (**Fig. S2B**).

(11) Lines 156-157: provide a reference to “recently developed proximity labeling methods”

We have added the following references to this statement:

- Lee, S.-Y. et al. Engineered allostery in light-regulated LOV-Turbo enables precise spatiotemporal control of proximity labeling in living cells. *Nat Methods* 20, 908–917 (2023).

- Zhu, H. et al., Tyrosinase-Based Proximity Labeling in Living Cells and In Vivo. *J. Am. Chem. Soc.* 146, 7515-7523 (2024).

(12) Line 376: what could be the relevance of identifying putative transcription factors in the cilium?

We have rephrased the sentence to explain the relevance of identifying putative transcription factors in the cilium to: “We further identified many signaling components, such as kinases and putative transcription factors that differed between cell lines and could mediate unknown ciliary signaling pathways.” These pathways have been speculated to exist, based on specific ciliopathy phenotypes.

(13) Lines 377-379 “explain clinical differences... to tackle this larger question in the cilia community”: It has already been shown that specialized cilia in different cell types have different proteomes, such as the photoreceptor sensory cilium (e.g. Liu et al., *Mol Cell Proteomics* 2007, PMID: 17494944). This should be included in this discussion: it is not a new concept.

We thank the reviewer for pointing this out. We are well aware that defects in specialized cilia types, such as photoreceptors or olfactory receptors, which have been studied for several decades, cause specific phenotypes. What we wanted to highlight was that proteomic differences in seemingly similar primary cilia, which are present on many different cell types, might explain why defects in particular genes cause different ciliopathy symptoms. We added the specialized cilia types and re-phrased this section to clarify our point of discussion.

(14) Line 321: typo: cel0l toxicity

This (and other) typos have been fixed in the revised version.

(15) western blotting/western blot should be without capitals (used with capitals throughout the text)

As the spelling of technology is often subject to journal style our original version included capital letters, which we have changed to lower case throughout the manuscript.

(16) It is not clearly visible what the colors in (B) mean in Figures S4 and S5. Please optimize this, perhaps by enlarging the representative part of the clustering graph and adding representative labels to make it comprehensive.

Thank you very much for this feedback. We have reworked the **Figures S4 and S5** by numbering and adding zoomed views onto the relevant clusters, which we hope solved this issue.

Reviewer #3 (Remarks to the Author):

This manuscript by Sroka et al. describes a clever adaptation to classical APEX2-based proximity labeling. By co-localizing APEX2 with the D-amino acid oxidase (DAAO) enzyme in a targeted subcellular region, the authors elegantly generate hydrogen peroxide in situ, circumventing the widespread off-target labelling and cytotoxicity usually observed when H₂O₂ is added exogenously. They term this approach “iAPEX,” and demonstrate that it greatly improves specificity and still retains the high temporal resolution typical of the APEX method. They further show how iAPEX works in multiple mammalian cell lines that have historically shown high background with classic APEX, and also provide a nice proof-of-concept in *Xenopus laevis*.

I see this as a significant advance for proximity labeling, especially if one aims to analyze fine structures (like primary cilia) in cell lines or tissues that are not well-suited to external H₂O₂ addition. The experimental data presented, immunofluorescence, expansion microscopy, Seahorse assays, plus TMT-based proteomics, gives strong support to the claims. The manuscript is clearly structured, well-illustrated, and of broad interest to cell biologists and proteomic researchers.

Nevertheless, below I have some suggestions that should further strengthen the final version.

Major comments

1. Kinetic range and concentration considerations

The authors show a thorough characterization with different D-amino acids (D-Met, D-Ala, etc). It is good if they highlight explicitly how widely this approach can be adapted in other cell contexts. Possibly, in some cell types, the amino acid transport or local oxygen availability might limit the H₂O₂ generation. A bit more discussion here would be helpful so that reader's from other fields are not uncertain how best to adapt it.

We thank the reviewer for pointing out this shortcoming of the previous version of the manuscript. We have expanded the discussion on potential cell type-specific differences of amino acid transporters, oxygen availability and redox systems that may affect application of the iAPEX system.

Moreover, we have added more experimental data for other applications on different organelles, which highlights the importance of time scales and substrate concentrations (see **new Fig. 6** and responses to reviewers #1 and #2). We have also generated more data on the in vivo application in *Xenopus laevis*, which we present in a re-worked **Fig. 7** and **new Fig. S7**. Our new data exemplifies some adaptations that were required to optimize the labeling conditions in the specific contexts, which we hope will help readers from other fields to successfully establish similar workflows.

2. Comparison to related proximity-labeling methods

The authors nicely mention that iAPEX addresses the major limitation of classic APEX. They also reference other new strategies, such as split-APEX, TurboID, or TransitID. However, short paragraph in the Discussion that positions iAPEX in direct comparison (in terms of timescales, background labelling, substrate costs, etc.) would be beneficial for us specialists.

We were happy to expand the discussion for a better comparison of the available technologies. Moreover, we have added additional analyses for a comparison of the iAPEX technology with APEX2 and TurboID in the context of primary cilia in the **new Figs. S6A-D** (see response to reviewer #2).

3. In vivo feasibility

The demonstration of iAPEX in *Xenopus laevis* is nice and quite convincing. It would be interesting if the authors expand slightly on potential complexities in adult tissues or in organs with lower oxygen tension, or mention if an optimized DAAO variant might be needed in future.

We thank the reviewer for another very valuable comment. We have expanded the discussion on the mDAAO enzyme (ref. 60/61) to include knowledge on oxygen availability in tissues and highlighted the the potential impact of the mDAAO enzyme on oxygen availability and toxicity. Moreover, we have added a new paragraph on the limitations and possibilities for in vivo applications (see responses to other reviewers).

4. Replicate variation and proteome-scale reproducibility

While the authors do show replicates, it might reassure readers to see more detail on replicates correlations (like Pearson's r) or a mention how stable is the identification of cilia proteins across replicates. Even short clarifications or a reference to the cluster correlation in the supplement would be nice for completeness.

We have added pairwise multiscatter plots and Pearson's coefficients for all replicates from the data on IMCD3 cells (Table S1) and NIH/3T3 cells (Table S2) in the **new supplementary Figures, Fig. S3B** and **Fig. S5B**, respectively. The data shows robust identification across all replicates.

5. Novel cilia candidates

Some of the newly identified cilia-localized proteins (for instance, PSKH1 or FCAP71/33) might open new directions in cilia biology. A brief speculation on how they could function or what domain features they carry may be interesting for the cilia community.

We have added another short section to discuss the newly identified and confirmed cilia-localized proteins with a specific focus on the apparently cell type-specific ciliary microtubule-associated proteins (see response to reviewer #1).

Minor comments

- Some phrasing could be toned down or rearranged for conciseness in the main figures, as they contain multiple panels. Also, clarifying the color-coding (like green for APEX2, red for biotin, etc.) more explicitly in the figure legends might help.

We have tried to rearrange phrasings for conciseness and explained color-coding more explicitly in the figure legends.

- For the Seahorse-based measurements, it would be helpful to mention approximate cell densities in the figure legends, so readers can replicate the oxygen consumption assays more precisely.

We have added cell numbers including coefficients of variation into the figure legend of the seahorse-based oxygen consumption measurements (**Figs. 2F-H**).

- In the text, “non-specific labeling” vs “off-target labeling” are used somewhat interchangeably. Perhaps unify the terminology in the final revision.

We thank the reviewer for this comment and have unified our phrasing to “non-specific” labeling.

Reviewer #4 (Remarks to the Author):

The manuscript by Sroka et al. presents a groundbreaking advancement in proximity labeling technology based on APEX2. This technique was originally developed by Alice Ting’s group at MIT, Boston, for mitochondria and other organelles. The authors have now successfully adapted it to primary cilia. One of the long-standing limitations of the classical APEX2 approach has been the specificity of labeling—partly due to potential mislocalization of the APEX enzyme, which in the traditional method is activated by externally added biotin-phenol and hydrogen peroxide (H₂O₂).

This manuscript addresses this limitation in a highly elegant manner by eliminating the need for exogenous H₂O₂ altogether. Instead, the authors employ local production of H₂O₂ through the targeted expression of a D-amino acid oxidase (DAAO) within the cilium itself. They refer to this refined approach as *in situ* APEX (iAPEX). This clever strategy increases labeling specificity, as both the APEX-tagged target protein and the H₂O₂-generating DAAO must be present at the same subcellular site—here, the cilium—for effective biotinylation to occur. And it allows to use APEX2 in cells that were not accessible to the method before due to massive background peroxidase activity.

Cilia pose a particular challenge in many respects: each cell typically forms only a single primary cilium, making the available material extremely limited. At the same time, primary cilia are at the center of a rapidly growing field of research, particularly in the context of ciliopathies—a group of genetic disorders caused by defects in ciliary structure or function, for which therapeutic options are still largely lacking.

From my perspective, this manuscript provides an elegant and innovative solution that allows for a more refined analysis of cilia and ciliopathies. I find the approach highly compelling, especially given its relevance to an important and currently under-addressed medical field. Preliminary data were already

presented at the 2024 Cilia Meeting in Dublin, where they generated substantial excitement, and are available as preprint.

We thank this expert cilia reviewer for their kind words and assessment of our work.

That said, there are several points that the authors should address:

1. Temporal Resolution: The manuscript states a 5-minute labeling time (line 156), which contrasts with the commonly used 1-minute H₂O₂ pulse applied in previous studies, including those by Mick et al. A more nuanced discussion of the temporal resolution would be appreciated. I would also recommend softening the language in the abstract, where the phrase “live cell proteomics” may be somewhat overstated.

We have re-phrased both the abstract and the main text for a more differentiated discussion on the temporal resolution of the iAPEX technology. While we recommend 5 min labeling times for cilia applications, shorter labeling times are still possible and in some applications even advised (see 2 min labeling for the lipid droplet application in the **new Fig. 6D**).

2. Validation of Ciliary Localization: The validations using the ALFA tag appear relatively weak. To convincingly demonstrate the specificity of the anti-ALFA staining, negative controls should be included.

We thank the reviewer for noting this weakness in the previous version of our manuscript. We now present new data to expand our analysis of potential candidate cilia proteins (see also response to reviewer #2). The new experiments to validate the localization of candidates have now been performed after generating stable cell lines expressing the transgenes for a better assessment (see **new Fig. 5F**). The new data also includes a negative control, as suggested by the reviewer. During the course of these experiments the previously weak localization of MINDY3 in cilia could not be confirmed, such that we have removed this protein from the figure. While we could not confirm the localization of MINDY3 (and other proteins) in primary cilia by expressing tagged versions or using available antibodies, we do not interpret this as the proteins being absent from cilia (absence of proof is no proof of absence).

In addition, we analyzed more candidates also in NIH/3T3 cells to confirm the cell type specificity of some candidates (as requested by reviewer #1), which we are presenting in the **new Fig. 5G**. Also here, we have tested more candidates without much success, however, final assessment for each candidate would require the generation of new tools, such as specific and confirmed antibodies -a task for the entire cilia community.

3. In Vivo Application: It would be exciting to see whether iAPEX can be applied in vivo. Is there a specific reason why biotin-phenol/D-Norvaline treatment was performed post-fixation? Clarification and discussion of this point would be helpful.

We thank the reviewer for his excitement and are happy to clarify the reasoning for how the experiment was performed. As reviewer #3 pointed out, one major challenge for the application of the technology is the availability of the DAAO substrates O₂ and D-AAAs, which appears particularly challenging in vivo. In the previous version of the manuscript, we therefore decided to include data showing the labeling post-fixation, which 1) highlights that the enzymes remain active after a fixation with paraformaldehyde, and 2) resulted in more robust labeling across the embryos. We have now moved this data into the **new Fig. S7**.

We attribute the previous variability to limited substrate availability and have extended the discussion on AA transporters and oxygen availability (in response to reviewers #1 and #3). Nonetheless, we are happy that we have succeeded in optimizing the in vivo labeling procedure using D-AA in *Xenopus laevis* embryos, such that we could replace the post-fixation data with data after in vivo iAPEX labeling, which we present in **Figs. 7K-L** (and controls in the **new Fig. S7D-E**).

As suggested by other reviewers, we have added a short paragraph to the discussion on substrate availability and include also some clarification on this point added by the reviewer. We have also expanded the discussion on challenges and possibilities for future in vivo applications.

4. Statistical Analysis: Statistical analyses should be included for the data presented in Figure 2.

We have added statistical analyses, including full description to **Figs. 2, S2D** and the **new Fig. S7A** (see also reviewer #1).

5. To better interpret the findings presented in Figure 1a, additional imaging parameters and methodological details would be helpful.

We have added the methodological details and imaging parameters to the figure legend of **Fig. 1A** to allow for a better interpretation of the presented findings.

6. The authors report APEX2 and DAAO localization at the ciliary membrane in Fig.1, even though NPHP3 n-term was used as the targeting motif. Could this reflect an artifact related to expansion microscopy? Co-staining with established ciliary membrane markers, in addition to acetylated tubulin, could help clarify this point.

We indeed used the N-term of NPHP3 for targeting both enzymes to primary cilia. The N-term of NPHP3 includes a myristoylation motif, which is integrated into the ciliary membrane via this fatty acid (Wright et al., 2011, Mick et al., 2015). Hence, the constructs do not co-localize with the axonemal marker acetylated tubulin, as seen in expansion microscopy (**Fig. 2A-B**). We have now included line plots (as suggested by reviewer #2) to better assess the precise localization of the NPHP3-targeted transgenes (**new Figs. S2A-B**).